# Seasonal and Diurnal Surface Temperatures of Urban Landscape Elements

**Jane Loveday [1],\* , Grant Loveday [2], Joshua J. Byrne [1], Boon-lay Ong [3] and Gregory M. Morrison [1]**

1   Curtin University Sustainability Policy Institute, School of Design and the Built Environment, Curtin University, Bentley 6102, Australia; josh@joshbyrne.com.au (J.J.B.); greg.morrison@curtin.edu.au (G.M.M.)
2   Independent Researcher; Booragoon 6154, Australia; grant.k.loveday@gmail.com
3   School of Design and the Built Environment, Curtin University, Bentley 6102, Australia; boon.ong@curtin.edu.au
*   Correspondence: jane.loveday@postgrad.curtin.edu.au

**Abstract:** In the context of the Urban Heat Island effect, landscape professionals need practical guidance to design for managing surface urban heat. The apparent surface temperatures of samples of 19 hard and soft landscape elements (LEs) found in Perth (Australia) were measured. Thermal images of LE samples on an oval were taken at a 1 m height. The study was conducted in two phases. Phase 1 LE surface temperatures relative to ambient ($\Delta T$) were measured over one day in all four seasons. LEs were ranked by average $\Delta T$, and maintained a similar order across seasons, with summer LEs the hottest. Some LEs were 30–44 °C above ambient in spring and autumn, so these seasons are also significant. Phase 2 repeated the summer test, but used only 14 larger LEs, which were well-coupled to the ground, i.e. more representative of in situ LEs. $\Delta T$ values were averaged over daytime and evening periods. Larger LEs were generally hotter than corresponding smaller LEs, with the effect more evident for heavier, denser LEs in the evenings. Future tests should be performed as per phase 2. Averaged measured values of grey pavers were the hottest, whilst ground-cover plants were the coolest. In the evening, grey pavers were also the hottest, whilst decking, soil and turf grass were the coolest. This data will help landscape professionals to assess and compare the thermal performance of different landscape designs, particularly when considering the time of use.

**Keywords:** Residential landscape; urban heat; landscape elements; surface temperature; thermal imaging

---

## 1. Introduction

The Urban Heat Island (UHI) effect is a complex phenomenon whereby ambient night-time temperatures in the urban canopy layer are typically much warmer than in rural areas [3]. However daytime UHIs also exist and their effect varies temporally [4,5], making it difficult to assess particular causes and effects. Temperature differences of between 0.4 and 11 °C have been reported by Santamouris [6] in a study of 101 cities across Asia and Australia. These hotter urban temperatures are caused by a decrease in pervious surfaces and vegetation (resulting in less evapotranspiration), an increase in anthropogenic heat (for example from industrial processes, transportation, and heating and cooling), the geometry of urban spaces limiting re-radiation and cooling breezes, and a general decrease in surface albedo through the use of darker surface materials.

A number of options for heat mitigation have been considered, including increasing green infrastructure, and altering the types and material properties of both pavement and roofing [7–12], *etc*. Within urban areas, pavement types, also referred to as surface treatments and defined here as

landscape elements (LEs), are the main driver of the surface UHI which represents the higher urban surface temperatures compared with pre-urban natural or rural surfaces [13]. Higher urban surface temperatures deliver more heat into the air through convection, thus also increasing the ambient UHI effect. Without shade, the heat from pavements or different LEs is highly dependent on the incident solar radiation. Other important parameters are wind speed, available moisture and LE material properties such as albedo, thermal conductivity, emissivity, and heat storage capacity [3].

Typically, more than 60% of urban surfaces are comprised of hard manufactured materials such as roof materials and paved surfaces [8]. Most studies of LE surface temperatures have examined asphalt and/or concrete which make up the greater proportion of the paved surfaces in urban public spaces [14–24]. The studies have generally concluded that asphalt, being of a darker colour (lower albedo), absorbs more solar radiation than concrete and thus has a higher surface temperature.

There have been significantly fewer surface temperature studies on other hard surfaces such as stone, ceramic and gravel [25–28], and on softer or pervious LEs such as mulch, sand, soils, turf grass or artificial turf grass, which are common features of domestic gardens [29–32]. Results for hard LEs, which demonstrated that smooth light-coloured materials were cooler whilst rougher, darker materials were warmer [27], were also applicable to soft LEs.

Previous studies have ranged in scope from measuring one up to 93 LEs, and only two of these studies have measured over the full annual seasonal cycle [15] measured temperatures and [20] measured the radiation balance) with the others measuring in summer only. The availability of instrumentation and the longevity of the study seemed to be limiting factors. In the warm temperate climate of Perth, Western Australia, autumn and spring may also give rise to increased urban heat, with average 10-year maximum ambient temperatures of 26.6 °C and 24.3 °C, respectively [33], even more so as the climate warms. This led to the testing of a comprehensive suite of typical residential and urban LEs (hard and soft) over four seasons in phase 1 of this study.

There have been a number of different measurement methodologies used previously. Some LEs have been measured in situ, with surrounding buildings or other LEs and weather conditions possibly confounding the results. In situ measurements have the advantage of being more representative of the true situation however comparing the effects of different LEs becomes difficult. Samples of LEs have been tested in the laboratory under controlled conditions, resulting in more comparable results, however solar radiation has to be simulated and LEs are insulated from the ground, thus not replicating the effects of heat exchange processes. Other samples of LEs have been studied in the field, and these have ranged in size from 120 mm x 120 mm [27], up to 30 m x 30 m [29]. Some of these LEs were insulated from the ground [14,25,27], whilst others were effectively coupled to the ground using sand, soil or a gravel bed as they would be in situ [15,29]. In phase 2 of this study, LE samples are placed in the field, and the effect of their size, and how effectively they are coupled to the ground is investigated.

In order to compare the diurnal surface temperatures of different LEs, a temporal method of separating the data is proposed in phase 2, and LEs are ranked according to their effect on urban heat. The intent of this study then, is to provide more detailed information to landscape designers for the assessment and comparison of the surface heat performance of different landscape designs. The purpose of this paper, although based in science, is not to precisely model the behaviour of the LEs, but more to present the most relevant data pertaining to management of urban heat through surface temperatures. This study was undertaken in a suburb of Perth, Western Australia. Perth's climate is classified as Csa under the Köppen–Geiger climate classification system [34]. Results are applicable to similar regions under comparable weather conditions for LEs measured in open areas.

## 2. Materials and Methods

Phase 1 was a test of the apparent temperature of different LEs across seasons. Phase 2 was a test of larger more effectively coupled LEs in summer. Some elements of phase 1 and phase 2 were the same, but some were different. Where differences occurred, it has been described in the text below.

### 2.1. Study Design—Phase 1

#### 2.1.1. Materials

Nineteen different LEs were tested in this phase. Both a location plan and a list of the LEs are shown in Figure 1. The layout of these elements ensured that none would shade any other, and that radiant or reflected energy from each would have minimal effect on any other. Most had a top surface area of at least 0.13 m$^2$. The limestone block had dimensions of $90 \times 160 \times 500$ mm; grey and sandstone coloured concrete pavers were each $40 \times 400 \times 400$ mm; red pavers consisted of 4 to 6 pavers of size $50 \times 150 \times 230$ mm placed together; sand and soil depths were 50 mm; pine bark mulch depth was 80 mm. The decking was treated pine (with no finish) $19 \times 1070 \times 360$ mm and raised 70 mm off the ground. LEs were either placed directly onto the turf grass (red, sandstone, and grey pavers; concrete, artificial turf grass, shade cloth, decking, and limestone), onto a polyethylene bag on the turf grass (white and polished black stones, and moist soil) or onto a polyethylene bag within a low-sided plastic crate (soil, asphalt, mulch, and white and grey sand). The crate allowed movement of samples whilst retaining the same surface area and depth of LE from season to season. The term turf grass describes grass which has been managed and irrigated, and in this study the turf grass was *Pennisetum clandestinum* (kikuyu). The three different coloured shade cloths were included in the study in order to understand the effect of colour on otherwise identical materials. They were not erected to provide shade but were folded twice (creating four layers) and placed on the ground in a similar manner to the other LEs.

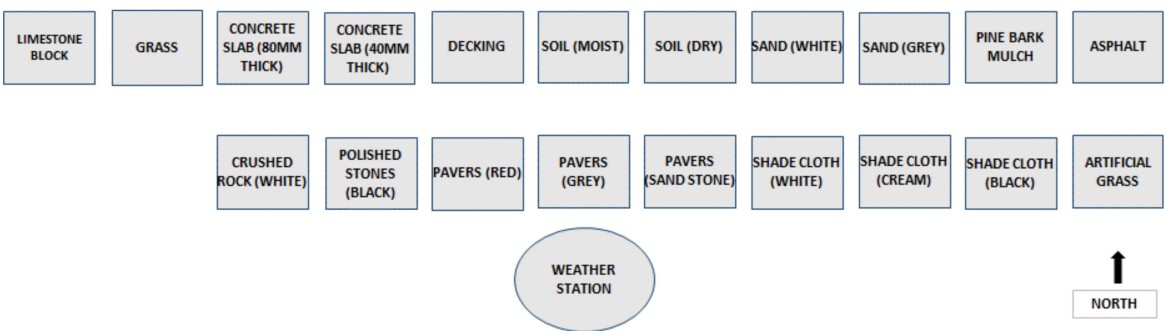

**Figure 1.** Plan of the landscape elements (LEs) placement on the oval for phase 1 (2017/8).

After the second season of data was collected (spring), the soil LE became dry. This change in moisture content would have a large effect on the surface temperature of future measurements and therefore, for each subsequent season, a new bag of the same soil was purchased and introduced as the moist soil sample, keeping the original soil sample as the dry soil sample. The term moist is defined then as the moisture level at which the soil is supplied from the landscaping company, however, this level was not quantified. Data collected from the soil sample in the first winter season was used for both the moist and dry samples, but in reality, the soil in winter is typically moist, as Perth has rainy winters and dry summers. Thus, the dry sample is representative for soil that has been left exposed to the elements.

#### 2.1.2. Field measurement Conditions

LEs were measured over one day and over each of the four seasons (the dates were: Winter—13 July 2016; spring—29 September 2016; autumn–9 April 2017; and summer—9 January 2018). The initial summer test was done in January 2017, however on this day ambient temperature reached 39 °C and this caused the thermal camera to malfunction. As a consequence, the summer test was repeated in 2018.

All test days had clear skies (no cloud cover), and ambient temperatures and wind conditions were similar to the average expected for that time of year. The LEs were placed on the Booragoon Primary School oval ($-32.035918°$, $115.826655°$), an open and turf grassed area (Figure 2). LEs were placed on

the oval the previous day to allow them to equilibrate before the tests began. The oval had a large sky view factor (SVF) which minimised the influence of possible thermal emitters or solar reflectors such as trees or buildings. A thermal imaging camera (Testo 876) with an accuracy of ±2 °C and a wavelength range of 8–14 μm provided images from directly above the LEs from a height of approximately 1 m. A thermal camera was used rather than a thermocouple or an infrared thermometer (which measures spot temperature), as this effectively measures multiple spot temperatures across the LE concurrently, taking into account any variation across the LE due to texture or uneven surface heating due to the suns position. Thermocouples are also difficult to attach to rough or natural surfaces. Half hourly images were taken all day, from a few hours before sunrise until a few hours after sunset. A portable weather station was set up on the oval to record air temperature, relative humidity, solar radiation, and wind speed and direction at a height of approximately 2 m. Safety tape was used to secure the site and prevent interference.

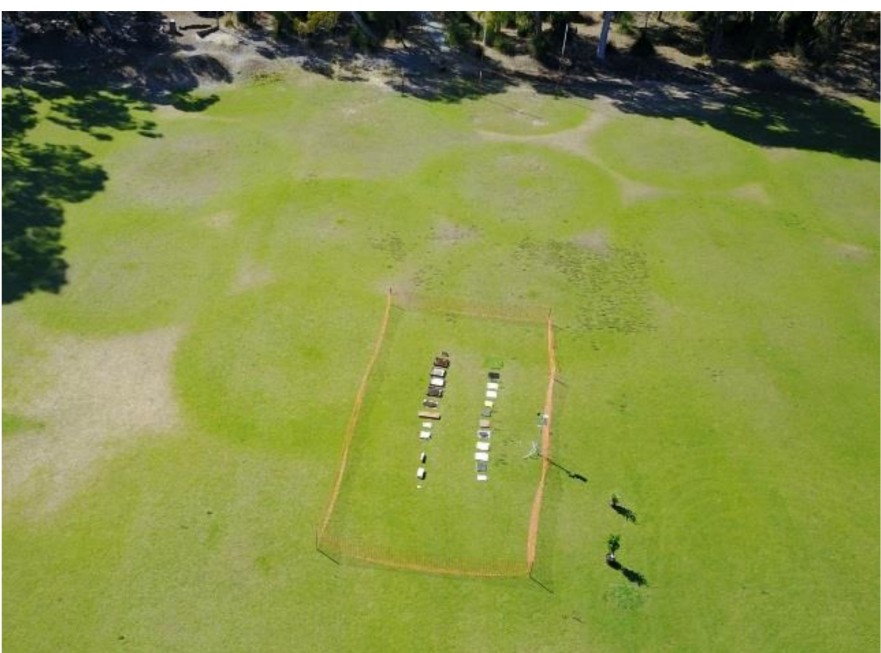

**Figure 2.** Aerial view of the oval test site showing the residential landscape elements with a large sky view factor.

A factor in phase 1 which may preclude the results from being more representative of a real situation, is the size of the LEs and the way they are coupled to the ground. The effect of a large number of connected concrete pavers for example, may be different from the effect of a single paver due to heat loss through the edges. The LEs were in contact with turf grass on the ground which is not how they would be installed in a garden i.e. in close contact with the underlying sand/soil. Both of these factors may change the amount of heat transferred away from the LE at different times of the day. To provide a better understanding of how this may impact the results, a second phase to the study was added.

*2.2. Study Design—Phase 2*

2.2.1. Materials

Phase 2 involved a subset of 14 of the same LEs (listed in Table 3), using identical materials where possible. The original artificial turf grass and timber decking materials could not be obtained, so products which were as similar as possible were used. These new materials, however, were both slightly darker in colour. The new artificial turf grass was synthetic Tuff Turf Multi with a 12 mm pile, and the new decking was a 19 mm hardwood (Merbau). All the LEs were of a larger size (1200 mm ×

1200 mm) and the depths of the soil, asphalt, mulch, and crushed white rock were 50 mm. Non-native annual groundcover (*petunias x hybrida*) and native seedlings (old man saltbush, *atriplex nummularia*) were also tested on this occasion in order to compare with turf grass. The petunias were in 100 mm pots closely packed together. The saltbush seedlings were in $50 \times 50$ mm containers and similarly closely packed together.

### 2.2.2. Field Measurement Conditions

Phase 2 used the same measurement methods as phase 1, but the LEs were larger and more effectively coupled to the ground. LEs were tested on one day in summer, on the 21 January 2019. Apart from the soil, all were coupled to the ground on a 50 mm packed layer of yellow sand (seen in Figure 3). The leaves of the petunias and saltbush gave nearly 100% coverage of the sand. Due to the difficult and time-consuming nature of constructing these larger samples (which were unable to be erected on a permanent site), only one seasons' worth of measurements were undertaken. The summer season was chosen, as LEs are expected to experience the highest surface temperatures at this time, resulting in higher levels of urban heat.

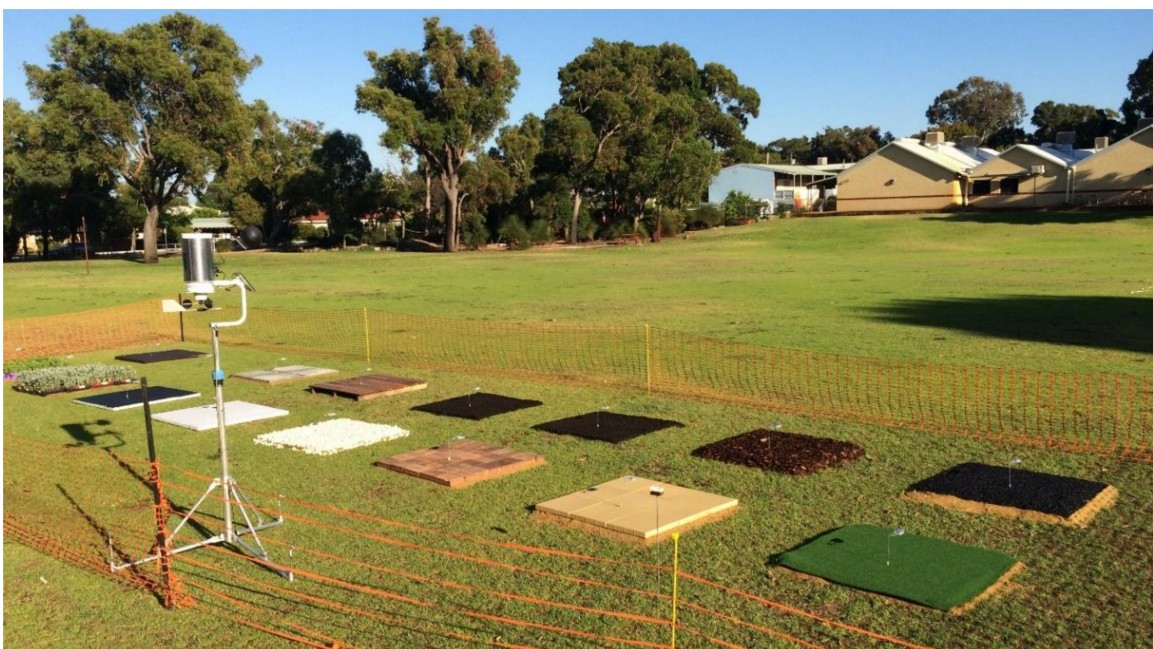

**Figure 3.** Larger LEs tested for phase 2 in Jan 2019, coupled to the ground with yellow sand.

### 2.3. Data Analysis

This section describes the method used in order to calculate the surface temperatures measured by the thermal camera, and gives the equations used for calculating the errors in these temperatures. It describes the choice of the parameters used for determining urban heat, and the temporal way these parameters were analysed.

### 2.3.1. Instrument Data

The manufacturer's software was used to analyse images from the thermal camera. The camera has two user inputs: the reflected temperature which represents the temperature of the surrounding environment or the reflected temperature ($T_{ref}$); and the emissivity ($\varepsilon$), which is a dimensionless number between 0 and 1, and is the ratio of the energy radiated from the surface compared to that radiated from a black body at the same temperature and wavelength.

If accurate values of the emissivity of the LE, and of $T_{ref}$ are known, the software converts the measured radiation into a thermodynamic temperature. However, if these values are only estimates,

this temperature is called an apparent surface temperature ($T_{app}$). The actual $\varepsilon$ and $T_{ref}$ are difficult to determine accurately in the field, and variations between the actual, and assumed values can lead to some error in $T_{app}$.

$T_{app}$ was determined for each LE by finding the average within a square drawn onto the central part of each LE image, approximately one third of the size of each LE (Figure 4).

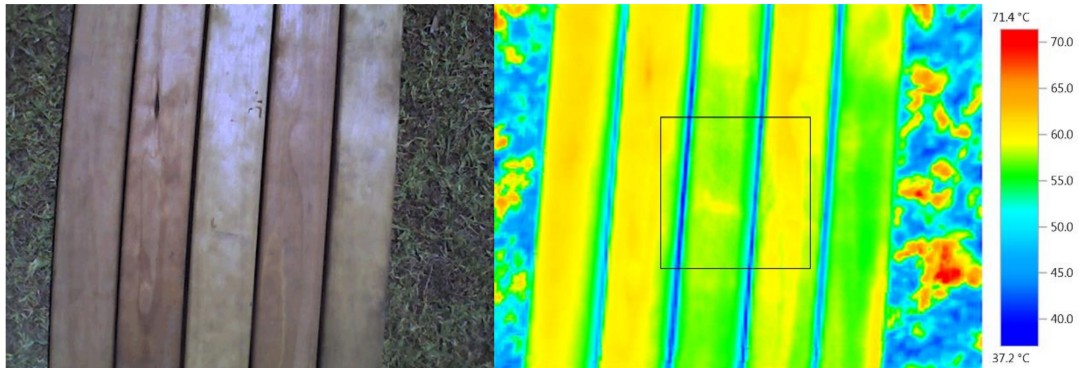

**Figure 4.** Real and thermal image of decking in summer, showing the averaging square which determined the apparent surface temperature of the LE.

The camera had been set to the value generally accepted for most surfaces of $\varepsilon = 0.95$ and an outside $T_{ref}$ of −30 °C. The actual value of $T_{ref}$ at each half hourly measurement period was found by setting $\varepsilon = 1$ and taking an image of crumpled aluminium foil (as per the reflector method in [35]. Using the camera software and the actual $T_{ref}$, a more accurate value of $T_{app}$ was found, although still using an $\varepsilon$ of 0.95.

### 2.3.2. Determining the Main Parameter for Urban Heat

The heat of a LE contributes to the ambient urban heat island *via* convection. The amount of heat added to the atmosphere is dependent on the difference between $T_{app}$ and ambient temperature $T_{amb}$. This is seen in Equation (1), derived from Newton's law of cooling for convection, where the heat flux due to convection $q_{conv}$ (W·m$^{-2}$) is:

$$q_{conv} = h_{conv}\Delta T \tag{1}$$

where $h_{conv}$ is the convection coefficient (W·m$^{-2}$·K$^{-1}$) and is dependent on both the material properties of the LE and the wind speed, and is assumed to be linear for wind speeds under 5 m·s$^{-1}$ at the surface [36]. Also, where:

$$\Delta T = T_{app} - T_{amb} \tag{2}$$

Hence, the heat convected into the air by each LE is proportional to the difference, $\Delta T$. The convective losses are not calculated in this study, rather Equation (1) is used to explain the process through which heat is added to the urban environment. The parameter $\Delta T$ was used to represent the effect of LEs on urban heat

### 2.3.3. Temporal Parameters

On each test day, measurements were ceased a few hours after sunset when it was observed that the LEs temperatures became close to or less than $T_{amb}$. When comparing different LEs, the overnight surface temperatures were thus expected to contribute very little to the averages calculated using the actual measured data. The 30 min $\Delta T$ data were averaged over the whole measurement period for each day of each season. This was denoted as $\Delta T_{av}$ and was representative of the total amount of heat each LE would be contributing to the urban environment.

Phase 2, summer data were separated into daytime and evening periods. Daytime was when the LEs were in the sun, and evening times were those during the two hours post sunset. Summer daytime

values were the averaged $\Delta T$ from 07:00 to 19:00 (*Day$\Delta T_{av}$*), and evening values from 19:30 to 21:00 (*Eve$\Delta T_{av}$*). This separation was done to investigate any effects of thermal mass, as there will be a greater distinction between different LEs over the evening period, when the effect of thermal mass becomes more significant. Not only does the UHI vary according to time of day [4], but the time of use of the urban space can also vary. Daytime surface temperatures may influence how the space surrounding the LEs is able to be used during the hotter periods of the day, whilst evening temperatures could impact the amount and speed of cooling required for buildings when people initially return home at the end of the day. This information would give landscape designers a better understanding of the temporal effect of LEs, and potentially give further insight into the effect of landscaping on the varying diurnal UHI effect. No previous literature on separating the data in this way was found.

Apart from summer 2018, phase 1 data were not separated into day and evening as the small size and low thermal coupling meant evening data were not significantly different from each other. Summer 2018 data were separated in order to compare with summer 2019 data.

### 2.3.4. Uncertainty in $T_{app}$

The incident radiation to the camera $M$ (W·m$^{-2}$) includes both the radiant exitance ($M_e$) emitted by the LE due to its temperature, and any reflected radiant exitance from the LE ($M_{ref}$). $M_{ref}$ is dependent on $T_{ref}$, and occurs because materials are not perfect absorbers or emitters (unlike a black body), meaning they have an $\varepsilon$ of less than 1.

The errors for different $\varepsilon$ and $T_{ref}$ at different surface temperatures are calculated using the general form of the Stefan–Boltzmann equation to determine $M_e$ as given in Equation (3):

$$M_e = \sigma \varepsilon T^4 \tag{3}$$

where $T$ is the absolute temperature of the body (K) and $\sigma$ is the Stefan–Boltzmann constant (5.67 × 10$^{-8}$ W·m$^{-2}$·K$^{-4}$).

$M_e$ was found using $T_{app}$ and the value of $\varepsilon$ used when the image was acquired for each LE. $M_{ref}$ was calculated using the values of (1– $\varepsilon$) and $T_{ref}$ which were used when the image was acquired. A third radiant exitance is that from the atmosphere, however because the distance from the camera to the LEs was only 1 m, this term is close to zero and can be ignored [37]. The two exitances were summed together to give the total radiant exitance received by the camera as shown in Equation (4).

$$M = M_e + M_{ref} = \sigma\varepsilon\left(T_{app} + 273.15\right)^4 + \sigma(1 - \varepsilon)\left(T_{ref} + 273.15\right)^4 \tag{4}$$

Equation (4) was re-arranged and used to determine different apparent temperatures for different values of $\varepsilon$ and $T_{ref}$ based on the assumption that most LEs have an $\varepsilon$ around 0.95. Literature values for $\varepsilon$ of the same LEs are quite varied and are dependent on the surface properties of the LE which can change with time and temperature. For example, a rougher or dirtier surface, caused by aging, will tend to increase the $\varepsilon$ [38], hence a range of 0.99 down to 0.9 is reasonable for the LEs tested here. An example of the types of errors which may occur from the assumption of $\varepsilon$ and $T_{ref}$ values if the actual values of $\varepsilon$ and $T_{ref.}$ were known, are shown in Table 1. Black and white shade cloths are used as the extreme values for winter and summer.

**Table 1.** Change in $T_{app}$ depending on changes in $\varepsilon$ and $T_{ref}$ from assumed values to hypothetical actual values for extreme temperature LEs.

| Example LE | $T_{ref}$ | $T_{ref}$ **Status** | Temperature (°C) | | | |
| --- | --- | --- | --- | --- | --- | --- |
| | | | $\varepsilon = 0.95$ (Assumed) | $\varepsilon = 0.99$ (Actual) | $\varepsilon = 0.90$ (Actual) | Maximum Difference |
| Black shade cloth | −6 | assumed | 85.0 | 82.5 | 88.4 | 3.4 |
| | −10 | actual | 85.1 | 82.5 | 88.6 | 3.5 |
| White shade cloth | −6 | assumed | 40.0 | 38.5 | 42.0 | 2.0 |
| | −10 | actual | 40.1 | 38.5 | 42.3 | 2.2 |
| Black shade cloth | −20 | assumed | 40.0 | 38.2 | 42.5 | 2.5 |
| | −30 | actual | 40.3 | 38.2 | 43.0 | 2.7 |
| White shade cloth | −20 | assumed | 10.0 | 9.0 | 11.4 | 1.4 |
| | −30 | actual | 10.4 | 9.0 | 12.1 | 1.7 |

From Table 1 we can see that a large change in $T_{ref}$ has a relatively small change in $T_{app}$ (<1 °C) whereas a change in $\varepsilon$ of 0.05 has a larger effect (up to 3.5 °C difference). Hence measurement inaccuracy in $T_{ref}$ is unlikely to significantly affect $T_{app}$, and the main error will be caused by an incorrect assumption of $\varepsilon$.

### 2.3.5. Uncertainty in Temporal Parameters

Assuming a non-normal distribution and using error propagation [39], the uncertainty in $\Delta T_{av}$ for $N$ measurement points was (Equation (5)):

$$Err\Delta T_{av} = \frac{1}{N}\sqrt{N(\Delta T_{app}^2 + \Delta T_{amb}^2)} \tag{5}$$

If the uncertainty in $T_{amb}$ was estimated to be ±0.5 °C and the uncertainty in each $T_{app}$ was ±1 °C (an estimation based on varying the location of the averaging boxes drawn onto the thermal images), then $Err\Delta T_{av}$ would be ±0.2 °C for each season. However, the maximum uncertainty in $T_{app}$ could be up to 3.5 °C because of the inaccuracy of the estimated emissivity (Table 1). The manufacturer's error for the camera of ±2.0 °C was assumed to be systematic and encompassed within the ±3.5 °C. This would give $Err\Delta T_{av}$ of ±0.7 °C for winter, and ±0.6 °C for spring, autumn, and summers 2018 and 2019. Similarly for measurements of $Day\Delta T_{av}$ data for both summers (25 measurement points) uncertainty is ±0.7 °C, and $Eve\Delta T_{av}$ data (4 measurement points), uncertainty is ±1.8 °C.

## 3. Results and Discussion

### 3.1. Weather Conditions

Weather conditions for each of the days measured were close to typical for that season according to data from the Bureau of Meteorology [40]. The conditions are shown in Table 2 and the temporal variation is shown in Figure 5. Due to the failure of the weather station that was used in winter and spring, data from the nearby Murdoch University weather station [41] which was 3.6 km away, was used for all seasons for consistency. The sensors used at this station were: solar radiation, a Middleton SK-01D silicon pyranometer; ambient temperature and relative humidity, a combined Rotronic 101A probe; and wind speed, a Synchrotac 706 propeller anemometer measuring at a height of 10 m. When the oval weather station was functioning, wind speed averages at 2 m above ground level were less than 60% of the Murdoch wind speeds. Wind speeds closer to the ground at the level of the LEs would be even less due to ground friction [42]. Higher wind speeds decrease the absolute value of $T_{app}$ by increasing convection.

**Table 2.** Weather conditions during the time of measurements over the test days.

| Season | Measurement Time Range | Air Temperature Range (°C) | Relative Humidity Range (%) | Max Solar Radiation (W·m$^{-2}$) | Average Daytime Tref (°C) | Wind Speed (m·s$^{-1}$) Average (std dev) |
|---|---|---|---|---|---|---|
| Winter 2016 | 06:00–20:00 | 0.6–16.5 | 28–98 | 605 | −25 * | 5.76 (1.68) |
| Spring 2016 | 05:00–20:30 | 7.0–18.7 | 24–81 | 941 | −25* | 5.24 (2.03) |
| Autumn 2017 | 05:00–21:00 | 12.0–27.4 | 25–82 | 831 | −17.5 | 1.51 (0.77) |
| Summer 2018 | 05:00–21:30 | 18.1–35.0 | 23–64 | 1049 | −7.2 | 2.70 (0.78) |
| Summer 2019 | 05:00–21:00 | 18.8–27.8 | 54–96 | 1009 | −7.8 | 1.97 (1.00) |

* This was the estimated value as the reflected temperature was below the limit of the thermal camera (<−20 °C).

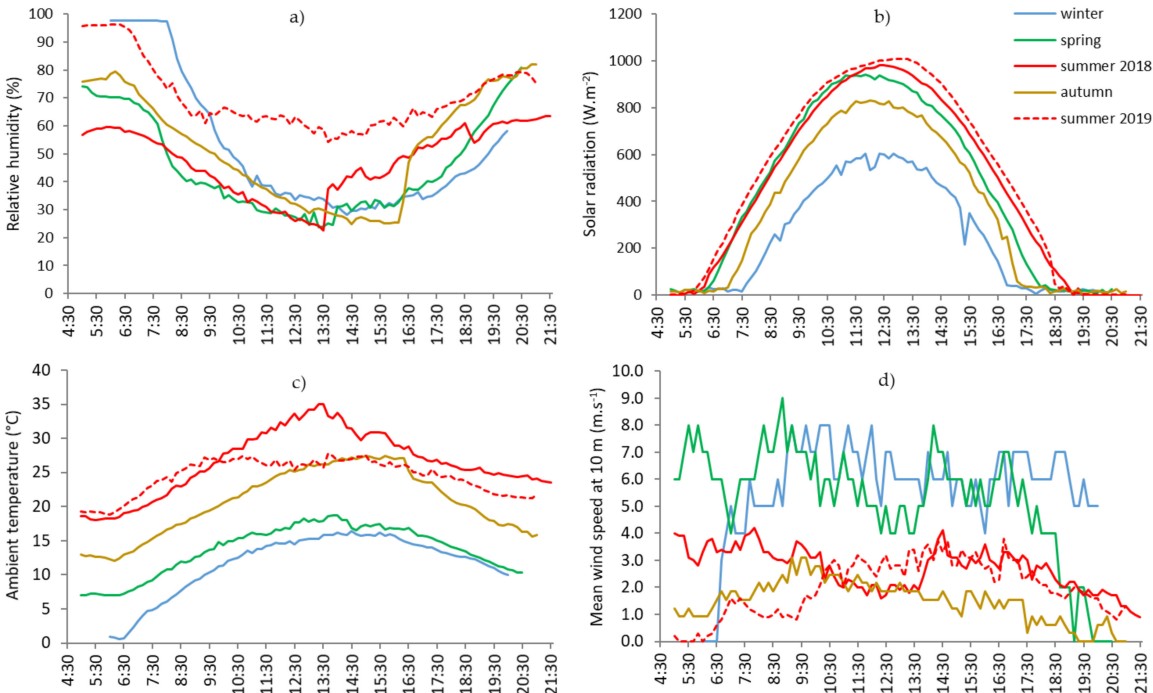

**Figure 5.** Seasonal summary of weather conditions over the measurement periods: (**a**) Relative humidity (%); (**b**) Solar radiation (W.m$^{-2}$); (**c**) Ambient temperature (°C); (**d**) Mean wind speed at 10m (m.s$^{-2}$).

### 3.2. Phase 1—Cross Seasonal Comparison

All surface temperature data, including phase 2 data, is given in Appendix B. Figure 6 shows a cross seasonal comparison of averaged 30 min apparent minus ambient temperatures calculated over the measurement period ($\Delta T_{av}$) at $\varepsilon = 0.95$ and at the measured $T_{ref}$ values. Data is tabulated in Appendix A. The negative values represent LEs which are on average below ambient temperature over the period.

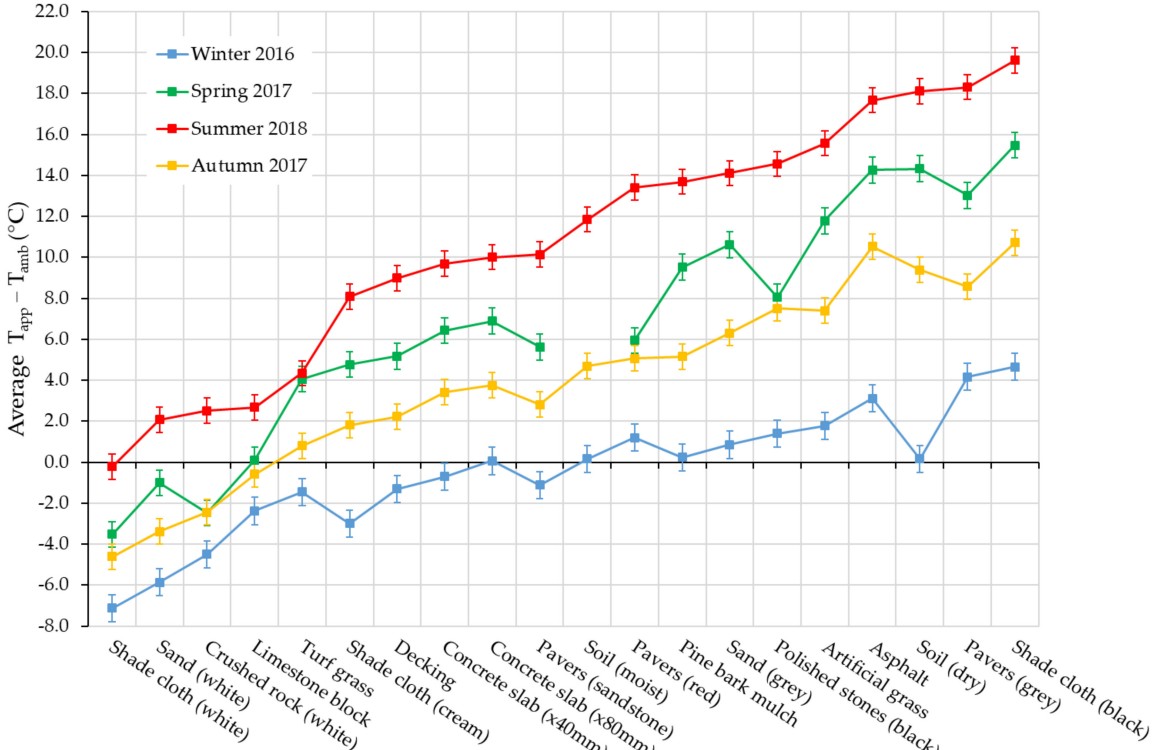

**Figure 6.** Seasonal $\Delta T_{av}$ (05:00–21:00) from each LE in phase 1 (°C). The LEs are ordered by summer values from lowest to highest. Error is ±0.7 °C for winter and ±0.6 °C for other seasons.

In general, Figure 6 shows that the lowest values of $\Delta T_{av}$ occurred in winter, and the highest in summer. Spring readings are slightly higher than autumn ones, which could be caused by several factors including different solar radiation, wind speed and $T_{ref}$. The spring data were not collected in the exact mid-season and the solar radiation was 110 W·m$^{-2}$ higher than the autumn value (Table 2). These results suggest that the temperature differences between the LEs and the ambient air are driven mainly by solar radiation. These results are similar to seasonal hard pavement temperatures measured by Li [15]. Unexpectedly however, the spring value for turf grass was similar to the summer value. This could be related to soil moisture content, as the watering schedule for the oval was unknown. Results showed the moist soil LE was around half the dry soil average in summer, confirming that moisture content, and hence evaporative cooling, also plays a significant role in surface heat. This has been shown by other researchers [8], and the cooling effect is particularly effective for moisture which is found in the top 25 mm of the LE [43].

The $\Delta T_{av}$ for all summer LEs were on average 4.1 °C (standard deviation 1.3 °C) higher than the next season of spring, not including the turf grass values. Even though summer surface temperatures were the highest, some LEs in spring and autumn also had high temperatures, with 6 LEs in spring, and 3 in autumn, registering maximums of over 30 °C above $T_{amb}$ (Appendix A). Average maximum ambient temperatures from 2008 to 2018, for spring and autumn, were 24.3 °C and 26.6 °C, respectively [33]. So, on some hotter spring and autumn days in Perth, LEs will have a negative effect on human thermal comfort, and may cause nearby buildings to require the use of energy for cooling.

Taking into account the error in the measurements of ±0.6 °C (Section 2.3.2), Figure 6 also shows that all white LEs and limestone are always similar to or less than $T_{amb}$ for all seasons except summer. Assuming there is no external addition of heat during the night, this means that over a 24 h period, these LEs would not be convecting heat into the atmosphere and would not be adding to the UHI effect. From visual observation, the limestone block, being an off-white or cream colour, does not appear to be as reflective of sunlight as the white LEs, however its $\Delta T_{av}$ is much closer to that of white sand and rocks, than to the cream shade cloth and sandstone pavers. It is worth noting that this LE is

a different shape, being more block-like, hence the results are not directly comparable with the other LEs. Summer black and darker coloured LEs (including asphalt, grey pavers, dry soil, and artificial turf grass), have the highest $\Delta T_{av}$, and are between 15 and 20 °C above $T_{amb}$ over the measurement period.

The effect of the colour of a LE can also be seen in Figure 6 when considering the three shade cloths (black, cream, and white) which were used to examine temperature differences between LEs of the same material type and varying only in colour. In summer, black is the hottest whilst white is the coolest, and cream lies in between, being 8.3 °C above white and 11.5 °C below black. Just under half of the incoming solar radiation is in the visible spectrum, so these results support the fact that the white shade cloth with a high albedo, can reflect most of this visible light, and thus remains much cooler than other colours. This effect can also be seen in the pavers (grey, red, and sandstone) with the darker colours being hotter. These trends are consistent with those found by Doulos, Santamouris and Livada [27] and Radhi, Assem and Sharples [25].

Despite similarity in colour however, the differences in thermal behaviour were notable for the artificial turf grass and the natural turf grass. Artificial turf grass was on average 11.2 °C hotter than turf grass in summer over the measurement period. Evapotranspiration is assumed to be the main cause of this difference, as well as the perviousness of the natural turf grass allowing any moisture from the soil to evaporate up though the surface, providing extra cooling. The artificial turf grass, consisting of a tightly woven plastic mat, does not allow significant moisture through from the soil and is thus likely to preclude any evaporative cooling.

Seasonal results show that summer has a significantly greater effect than the other seasons on increasing urban heat. However, as some LEs exhibit high surface temperatures in spring and autumn, these seasons must also be considered by landscape designers, even in a warm temperate climate.

### 3.3. Phase 2—Effect of LE Scale and Installation Method

This section examines how representative the results from the small LEs test would be of a more realistic situation, by comparing them with LEs that are larger, and more effectively coupled to the ground. The temporal $\Delta T$ data for the larger LEs are shown in Figure 7. The natural or pervious LEs reach $T_{amb}$ between 18:30–19:00, which is around sunset. The hard and dense LEs remain above ambient until around the last readings at 21:00. Despite being a low thermal mass product, artificial grass also only goes below ambient between 20:30 and 21:00, indicating its close ground coupling is increasing its thermal mass dramatically.

Data were also included for the two LEs of ground cover (non-native, petunias) and seedlings (native, saltbush) which were measured in Jan 2019 only, in order to compare these with turf grass. Doulos, Santamouris and Livada [27] separated surface temperature data into categories from 09:00 to 03:00, 11:00 to 15:00, and 22:00 to 03:00. Their 11:00 to 15:00 period was comparable to the $Day\Delta T_{av}$ period data in this study, however unfortunately, the critical evening period in the few hours just after sunset, was not captured specifically in their data categories.

Categorised data are shown in Table 3. The LEs are sorted by 21$^{st}$ January $\Delta T_{av}$ in order from lowest to highest.

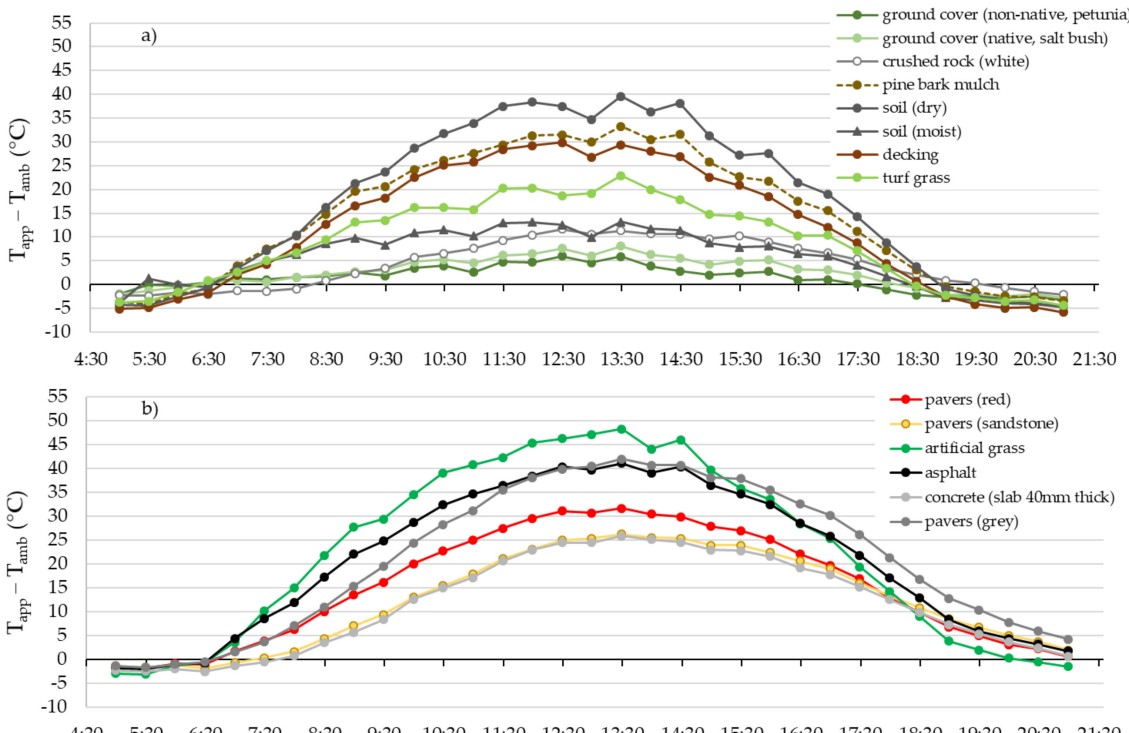

**Figure 7.** Apparent temperature minus ambient temperature for each LE on 21$^{st}$ Jan 2019: (**a**) soft and pervious LEs; (**b**) hard and manufactured LEs.

**Table 3.** Different sized and coupled LEs tested over two summers. Sorted by summer 2019 $\Delta T_{av}$.

| Average 30 min $T_{app}$—$T_{amb}$ (Emissivity = 0.95) Errors: $\Delta T_{av}$ ± 0.6, $Day\Delta T_{av}$ ± 0.7, $Eve\Delta T_{av}$ ± 1.8 °C | | | | | | | | |
|---|---|---|---|---|---|---|---|---|
| °C | 21st January 2019 (Large) | | | 9th January 2018 (Small) | | | Difference 2019–2018 | | |
| LE | $\Delta T_{av}$ | $Day\Delta T_{av}$ | $Eve\Delta T_{av}$ | $\Delta T_{av}$ | $Day\Delta T_{av}$ | $Eve\Delta T_{av}$ | $\Delta T_{av}$ | $Day\Delta T_{av}$ | $Eve\Delta T_{av}$ |
| Ground cover (non-native, petunia) | 1.3 | 2.2 | −2.7 | | | | | | |
| Seedlings (native, saltbush) | 2.4 | 3.7 | −2.3 | | | | | | |
| Crushed rock (white) | 4.2 | 6.1 | −1.0 | 2.5 | 5.0 | −3.8 | 1.7 | 1.1 | 2.8 |
| Soil (moist) | 5.4 | 7.9 | −4.0 | 11.8 | 17.7 | −3.7 | −6.4 | −9.8 | −0.3 |
| Turf grass | 8.7 | 12.3 | −3.5 | 4.3 | 7.4 | −4.5 | 4.3 | 5.0 | 1.0 |
| Decking * | 12.1 | 17.3 | −4.9 | 9.0 | 14.0 | −5.2 | 3.1 | 3.4 | 0.3 |
| Concrete (× 40mm) | 11.5 | 15.1 | 3.0 | 9.7 | 14.6 | −2.6 | 1.8 | 0.5 | 5.6 |
| Pavers (sandstone) | 12.4 | 15.9 | 4.3 | 10.1 | 14.9 | −1.1 | 2.2 | 1.0 | 5.4 |
| Pine bark mulch | 14.4 | 19.9 | −2.5 | 13.7 | 19.6 | −2.8 | 0.7 | 0.3 | 0.3 |
| Pavers (red) | 15.2 | 19.9 | 2.7 | 13.4 | 19.5 | −2.3 | 1.8 | 0.4 | 5.0 |
| Soil (dry) | 17.1 | 23.6 | −3.6 | 18.1 | 25.7 | −3.2 | −1.0 | −2.1 | −0.4 |
| Asphalt | 20.8 | 27.1 | 3.8 | 17.7 | 25.0 | −1.1 | 3.1 | 2.1 | 4.8 |
| Artificial turf grass * | 22.5 | 30.0 | 0.0 | 15.6 | 23.0 | −5.6 | 6.9 | 7.0 | 5.6 |
| Pavers (grey) | 21.0 | 26.8 | 7.1 | 18.3 | 25.6 | 0.8 | 2.7 | 1.2 | 6.2 |

*A darker product was used for the 21 Jan 2019 test.

Table 3 shows that except for pine bark mulch and dry soil, all LEs have different $\Delta T_{av}$ for the two summers (considering the error of ±0.6 °C). Apart from the soils, the larger and more effectively coupled LEs all have $\Delta T_{av}$ values up to 5.5 °C higher than the small LEs. To investigate when this difference is occurring, the data for daytime and evening was examined. Figure 8 shows the $Day\Delta T_{av}$ values, where the data have been sorted according to the 21 Jan 2019.

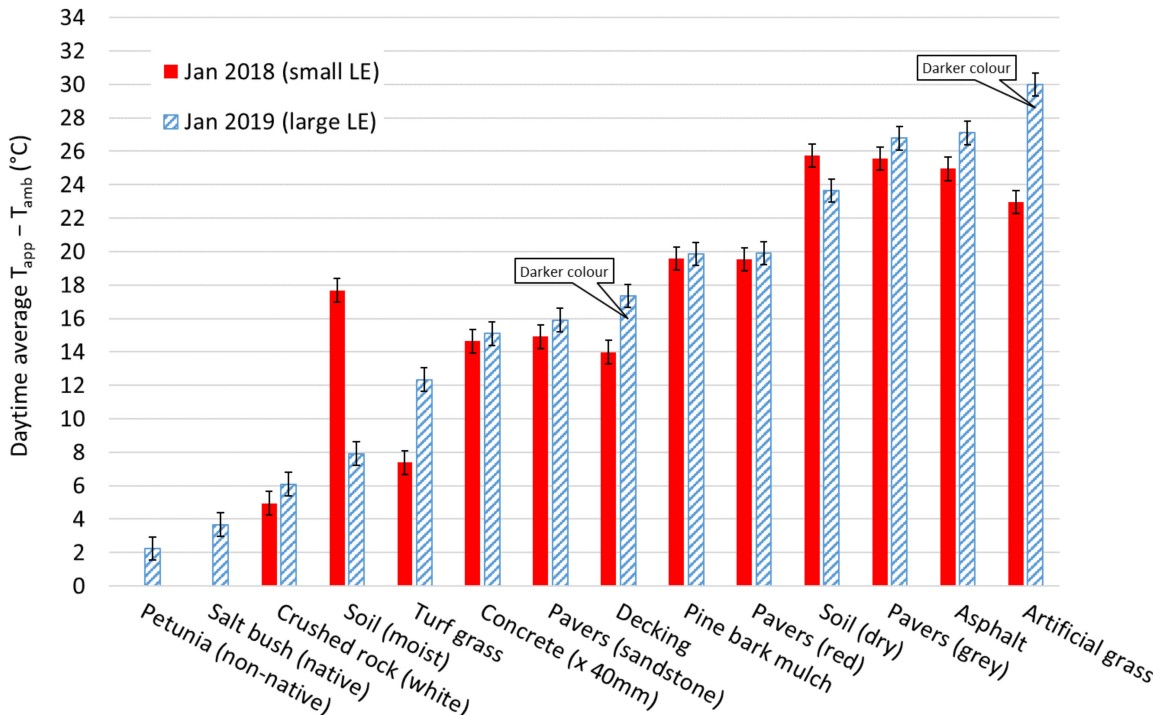

**Figure 8.** Comparison of the $Day\Delta T_{av}$ (°C) (07:00–19:00). LEs on 21st Jan 2019 were larger and coupled to the ground on a bed of sand. Results are sorted by the 2019 data set. (Note: Decking and artificial turf grass were of darker colour in 2019).

The LEs which were independent of size and coupling (within the error of ±0.7 °C) for $Day\Delta T_{av}$ were white rock, concrete (x40 mm), sandstone pavers, pine bark mulch, red pavers, and grey pavers. Doulos, Santamouris and Livada [27] found that the surface temperatures were the same for two different sized (30 cm x 30 cm, and 40 cm x 40 cm), groups of concrete tiles, insulated from the ground and measured over the summer daytime period from 09:00 to 18:00. These results agree with data from this study, however no other LE types were compared in this way.

The most obvious difference in $Day\Delta T_{av}$ is for moist soil. Whilst the moist soil sample was left untouched during the 2018 test, 9 L of water was added to this LE in the 2019 test on four occasions during the day. This was to broadly assess the effect of moisture. Although moisture levels were not measured, the addition of water more than halved $Day\Delta T_{av}$. Similarly, the moisture content of the turf grass was not measured, and the oval watering schedule was unknown. It is assumed from the data that the turf grass was watered more recently and/or thoroughly just before the 2018 test than the 2019 test.

The other obvious differences were the darker coloured artificial turf grass and darker coloured decking, which had higher $Day\Delta T_{av}$ values. The extent to which this difference was caused by the change in colour, or by the scale and coupling, cannot be determined. However, these results are included for reference. The reasons for the differences in the remaining LEs are discussed below in the evening data section.

A comparison between the daytime turf grass, and the petunias and the saltbush, shows the turf grass was warmer than the plants by 10.1 and 8.6 °C respectively. One reason for this is that the plant's potting soil may have a higher water carrying capacity than the sandy loam of the turf grass. Another consideration may be the apparent smaller leaf area index of turf grass, providing less shading and evapotranspiration, and hence less cooling.

Temporal data for asphalt $\Delta T$ are given in Figure 9. This demonstrates the effect over the day of increasing the size of the LEs and more effectively coupling them to the ground. The larger coupled LE in 2019 starts at a higher initial temperature and does not reach $T_{amb}$ before measurements finish at

around 21:00. In contrast, the smaller LE passes below ambient between 19:30 and 20:00. This may be due to either the losses though the edges of the smaller LEs, or the closer ground coupling of the larger LEs. Whilst there is only a 2.1 °C difference in $Day\Delta T_{av}$ results (Table 3), $Eve\Delta T_{av}$ data from 19:30 to 21:00 shows this effect more noticeably (Figure 10).

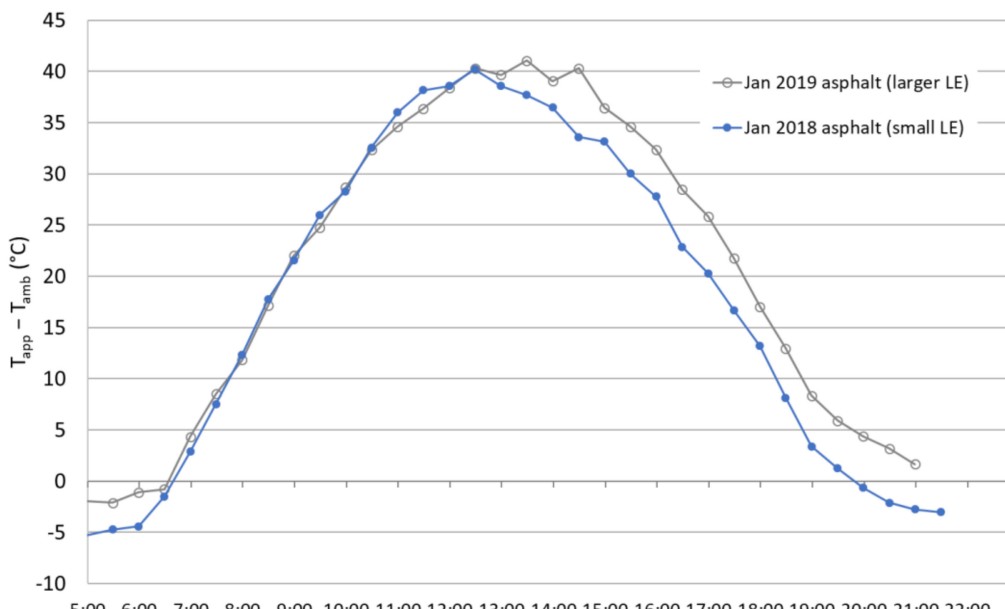

**Figure 9.** Temporal data for asphalt $T_{app}$—$T_{amb}$ for summers 2018 and 2019.

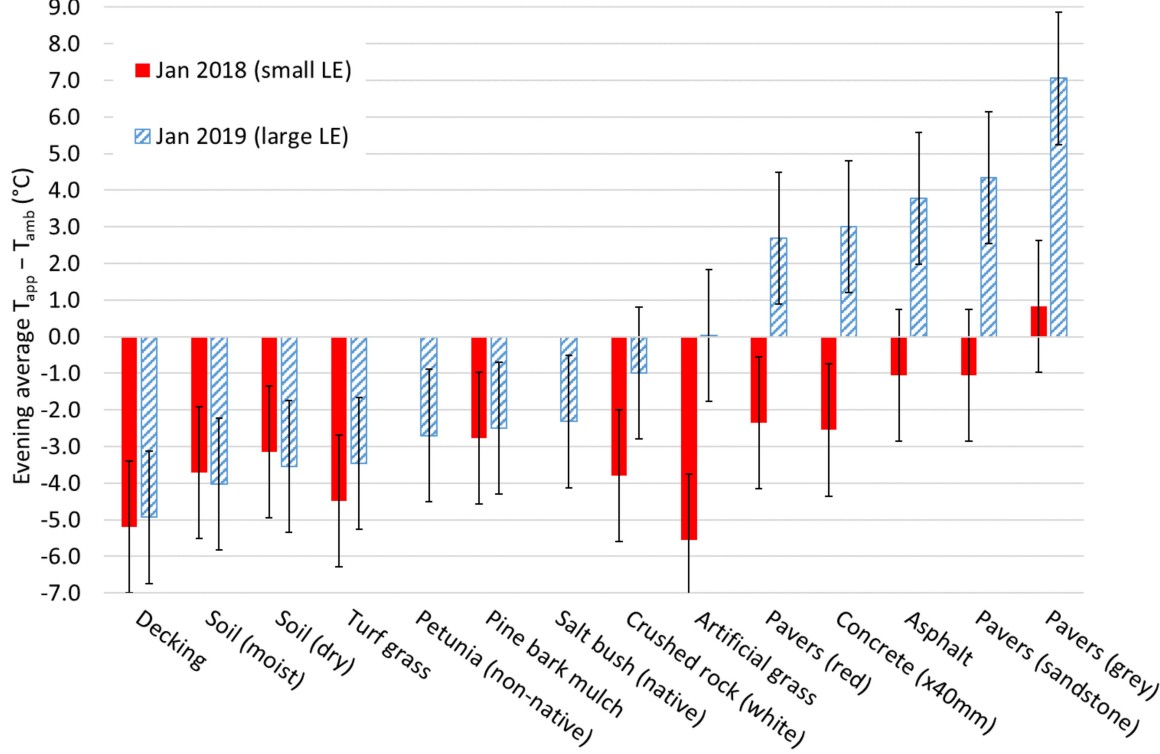

**Figure 10.** Comparison of the average 30 min $Eve\Delta T_{av}$ (°C) (19:30–21:00). LEs on 21 Jan 2019 were larger and coupled to the ground on a bed of sand. Data are sorted by the 21 Jan 2019.

Comparing 2018 and 2019 data from the values given in Table 3, $Eve\Delta T_{av}$ for decking, moist soil, dry soil, turf grass, pine bark mulch, and crushed rock (white) are all the same within the error of

±1.8 °C. Figure 10 is a good visual representation of this data, showing how the $Eve\Delta T_{av}$ for artificial grass, concrete, asphalt and all the pavers are markedly higher for the larger sized LEs. Three potential influencing factors for this are:

- Edge effects—the amount of heat lost through the edges of the LEs is proportionally greater for the smaller sized LEs;
- ground coupling—coupling to the ground by means of a bed of sand increases the thermal mass; and
- LE conductivity—internal air spaces (e.g. mulch) result in the LE being poorly conductive.

These are discussed in turn below.

Considering edge effects, the thermal images of concrete and grey pavers were examined and the smaller LE samples showed a cooling off around the edges in the evening. However, for the same sized image of the larger LEs, there was no temperature gradient visible (the cooler edges were out of the field of view of the camera). This was not the case for the more porous LEs of pine bark mulch and soil, where no edge effect was seen meaning they were losing heat equally across their surface regardless of surface area. Hence edge effects are a factor for dense solid LEs, but not for porous LEs.

The larger hard LEs are in close contact with the underlying sand and as such, provide a good pathway for the transfer of heat, effectively increasing the thermal mass of the LE. Montague and Kjelgren [29] found asphalt and concrete conducted more energy to the underlying soil than pine bark mulch. Increased thermal mass means the LE will stay warmer for longer as the stored heat from the ground moves back through the LE. This is not the case for the smaller hard LEs, as sitting on the grass, they were not in direct contact with the soil and did not have the increased thermal mass. This explains the $Eve\Delta T_{av}$ values being much greater for the larger LEs compared with the smaller LEs.

The larger porous LEs, although in direct contact with the ground, contain multiple air spaces (behaving like good insulators), minimising the transfer of heat into and out of the ground. These LEs do not conduct very well either through their own mass, or into the underlying soil. So, both the larger and smaller sized porous LEs do not have an increased thermal mass and hence their $Eve\Delta T_{av}$ are similar. On the other hand, dense LEs show a slight effect of size and coupling during the day and a markedly more noticeable effect in the evening.

As mentioned previously, the density of the asphalt LE would be less than that found for a compacted road surface. With higher density there would be increased thermal mass and the $Eve\Delta T_{av}$ for asphalt would likely be much higher, perhaps comparable to grey pavers.

Because differences are evident between the phase 1 and phase 2 LEs, particularly during evening periods, and based on the conclusions from the seasonal data, which noted the significance of measuring in spring and autumn, it is recommended that further measurements on large and well coupled LEs be conducted in the spring and autumn seasons.

### 3.4. Ranking LEs

To provide clarity in the data, the LEs have been ranked according to their total, daytime, and evening values of $T_{app}-T_{amb}$ (Table 4). Summer data from the larger, well coupled LEs were chosen for this ranking due to their closer representation of LEs in situ. This table could be useful in landscape design, as for a given landscaping situation and usage pattern, it may be better or worse to include particular LEs. For example, in a residential garden, it may be preferable to include LEs which are close to $T_{amb}$ in the evenings for those hot summer nights when the residents are opening their windows to try to cool their house. Alternatively, the $\Delta T_{av}$ values may be more significant if the overall heat being convected into the atmosphere is of greater interest.

It is notable that the rankings of $\Delta T_{av}$ and $Day\Delta T_{av}$ are similar, however the $Eve\Delta T_{av}$ shows a greater order change and is more dependent on the thermal mass of the LEs. The order change may occur because of the large error in the evening data (see Figure 10), where there is no significant difference in the ranking order of decking, moist soil, dry soil, turf grass, petunias, pine bark mulch,

and salt bush seedlings. Similarly, there is no significant difference in ranking between red paver, concrete, asphalt, and sandstone pavers for the 2019 $Eve\Delta T_{av}$. Accuracy would be improved by increasing the number of images taken during the evening period. These rankings could help to explain some of the varying temporal UHI effect, which is dependent on the mix of urban surfaces as well as on other factors such as the urban form, precipitation, and anthropogenic heat sources.

**Table 4.** Order of LEs from coolest to hottest for different time periods in summer 2019.

| °C | $\Delta T_{av}$ (05:00—21:00) Error ±0.6 | | $Day\Delta T_{av}$ (07:00—19:00) Error ±0.7 | | $Eve\Delta T_{av}$ (19:30—21:00) Error ±1.8 | |
|---|---|---|---|---|---|---|
| coolest | Petunia (non-native) | 1.3 | Petunia (non-native) | 2.2 | Decking | −4.9 |
| | Salt bush (native) | 2.4 | Salt bush (native) | 3.7 | Soil (moist) | −4.0 |
| | Crushed rock (white) | 4.2 | Crushed rock (white) | 6.1 | Soil (dry) | −3.6 |
| | Soil (moist) | 5.4 | Soil (moist) | 7.9 | Turf grass | −3.5 |
| | Turf grass | 8.7 | Turf grass | 12.3 | Petunia (non-native) | −2.7 |
| | Decking | 12.1 | Concrete (×40 mm) | 15.1 | Pine bark mulch | −2.5 |
| | Concrete (×40 mm) | 11.5 | Pavers (sandstone) | 15.9 | Salt bush (native) | −2.3 |
| | Pavers (sandstone) | 12.4 | Decking | 17.3 | Crushed rock (white) | −1.0 |
| | Pine bark mulch | 14.4 | Pine bark mulch | 19.9 | Artificial grass | 0.0 |
| | Pavers (red) | 15.2 | Pavers (red) | 19.9 | Pavers (red) | 2.7 |
| | Soil (dry) | 17.1 | Soil (dry) | 23.6 | Concrete (x40mm) | 3.0 |
| | Asphalt | 20.8 | Pavers (grey) | 26.8 | Asphalt | 3.8 |
| | Artificial grass | 22.5 | Asphalt | 27.1 | Pavers (sandstone) | 4.3 |
| hottest | Pavers (grey) | 21.0 | Artificial grass | 30.0 | Pavers (grey) | 7.1 |

In general, as expected, darker coloured and dry LEs were hotter than lighter coloured or moist LEs for both the $\Delta T_{av}$ and the daytime periods. This is due to the albedo of the LEs, with low albedo (darker) LEs absorbing more solar radiation during the day and thus becoming hotter, and the converse for high albedo LEs. Moist LEs are cooler due to evaporative cooling. For the evening period albedo is not a factor, and the heavier and denser LEs were hotter than the natural and pervious LEs. This is because the higher thermal mass of the materials such as concrete, asphalt and pavers mean that daytime stored heat is released over a longer period of time once the sun goes down. Natural and pervious materials have less thermal mass and a higher specific surface area, which mean less stored heat is available for release and the heat is released more quickly.

*3.5. Maximum Temperatures*

Figure 11 shows a snapshot of the maximum surface temperatures and the maximum surface temperature above $T_{amb}$, for LEs measured in summer, 2018. These were calculated at $\varepsilon = 0.95$ and $T_{ref}$ = ~ −4 °C and error bars are based on an emissivity range of 0.95 to 0.90 (positive error) and 0.95 to 0.99 (negative error), the true emissivity being unknown. The higher the surface temperature, the more the LE will be adding to the UHI effect through convection. Dry and dark coloured LEs, and artificial turf grass are particularly hot being more than 30 °C above both $T_{amb}$ and above turf grass. These results are more extreme than measurements by Kuang, et al. [44] who found hard impervious surface were between 6 and 12 °C higher than plants or vegetated surfaces. However results for concrete and asphalt are similar to Li [15] who found the summer maximum for asphalt was around 20 °C higher than for concrete, and Doulos, Santamouris and Livada [27], who found a difference of 17 °C. White LEs and limestone are less than 10 °C above $T_{amb}$.

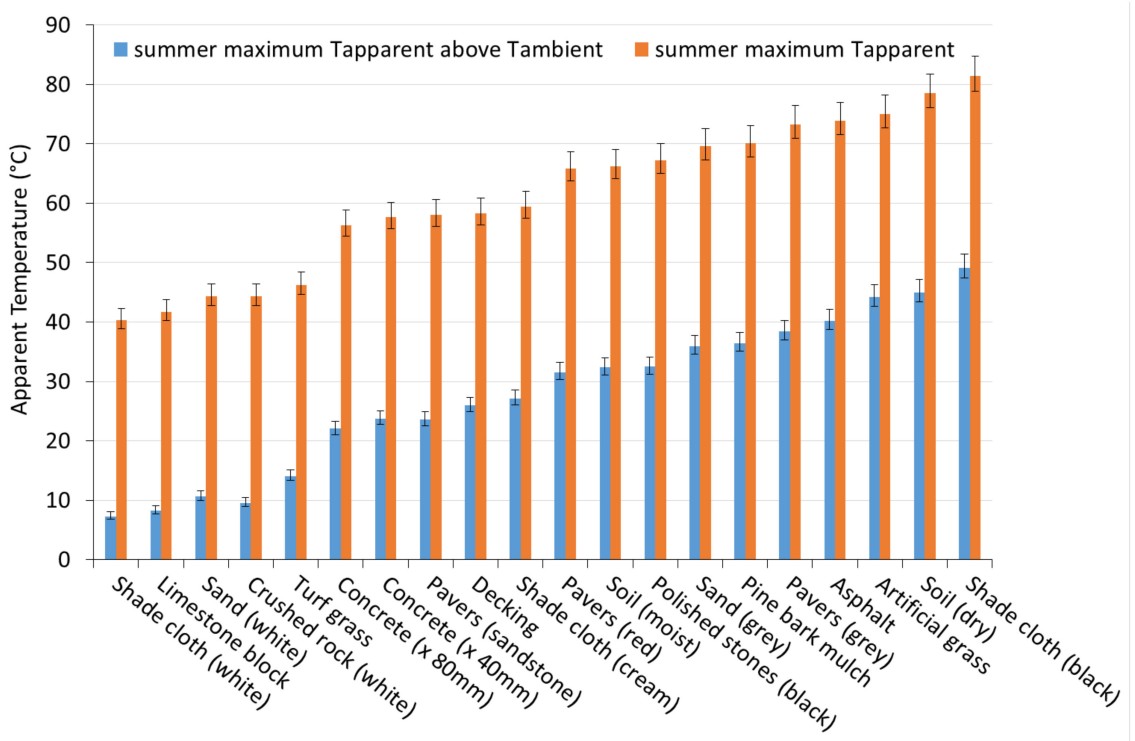

**Figure 11.** Maximum $T_{app}$ and $T_{app}$—$T_{amb}$ for all LEs in summer 2018 ($\varepsilon$=0.95, $T_{ref}$ = ~ −4 °C). Errors are based on emissivity potentially varying from 0.90 to 0.99.

### 3.6. Limitations

Using an $\varepsilon$ of 0.95 for all LEs will result in an error in $T_{app}$ depending on the temperature of the LE and $T_{ref}$ as discussed previously. Although this is a reasonably low error, caution must still be taken when comparing $T_{app}$ and $\Delta T$ results. If $\varepsilon$ were able to be accurately measured for each LE, this limitation would be overcome.

The moisture contents of the turf grass, sand, soil, and plant potting soil were not measured. This could have provided a better indication of the effect of evaporation on the energy balance.

The asphalt used in this study was not fully representative of a road surface as it could not be compacted down using heavy machinery. Thus, the density was less than asphalt in situ and the thermal mass relatively less.

Measurements were done over one day in each of the four seasons and as such, are representative only of days with similar solar radiation and sun inclination. Similarly, although wind speeds were calm to low, variations would affect surface temperature results across the day and between seasons. Unfortunately, collection of data is not always possible at optimum times. Applying this data to different days would require modelling which has not been performed in this publication.

### 4. Conclusions

The purpose of this research was to present the most relevant data pertaining to management of urban heat through surface temperatures. The results of this study are applicable to LEs in a high SVF area, under relatively calm wind conditions with no cloud cover, and in a warm temperature climate classified as Csa in the Köppen–Geiger climate classification system.

The difference between the apparent surface temperature of LEs and the ambient temperature ($T_{app}$–$T_{amb}$) is proportional to the amount of heat convected into the air which can add to urban heat. The additional heat from LEs can make air and surface temperatures even more uncomfortable when summer $T_{amb}$ are already high, hence summer is the most critical season to investigate. However, as some LEs exhibit high surface temperatures in spring and autumn, the effect of these seasons must

also be considered by landscape designers, even for a warm temperate climate. This appears to be a recommendation as yet unmentioned in the literature.

Darker and dry LEs will contribute more to urban heat through their higher surface temperatures and subsequent higher rates of convection. Both whiter LEs and plants will have a relatively smaller contribution as they remain close to ambient temperature across seasons.

Thermal mass and edge effects can affect the surface temperature of a LE. Therefore, future test work on samples of different LEs, particularly those with a high thermal mass, should use large and effectively coupled LEs, matching as close as possible to properly installed LEs. It is recommended that further measurements of these LEs be undertaken in spring and autumn.

Ranking typical urban and residential LEs by surface temperature is useful for landscape designers when designing for urban heat. The $\Delta T_{av}$ ranking is relevant for overall urban heat as it quantifies how much heat is convected into the atmosphere from each LE. The daytime ranking is important for landscapes where daytime use is prevalent, whilst the evening ranking is important for when people are trying to cool their homes in the evening. Previous literature on separating the data into these specific categories has not been found, but this method may be useful for understanding temporal UHI variations. Using these 3 ranking categories, landscape designers can select LEs which are appropriate for different usage patterns, and which will aid in the management of local urban heat. To expand this database, alternative LEs can be tested in a similar fashion alongside some of the existing LEs for comparison. A new database for LEs typical of other locations can be created *via* similar test work. Landscape professionals can use ranked apparent surface temperatures to assess the thermal performance of different landscape designs.

**Author Contributions:** Conceptualization of this research, J.J.B., B.-l.O., J.L. and G.L.; methodology, J.L. and G.L.; formal analysis, J.L. and G.L.; investigation, J.L.; data curation, J.L.; writing—original draft preparation, J.L.; writing—review and editing, J.L., G.L., J.J.B., B.-l.O and G.M.M.; visualization, J.L.; supervision, J.J.B., G.M.M. and B.-l.O.; project administration, J.L. and J.J.B.; funding acquisition, J.L. and J.J.B.

**Funding:** This research was supported by an Australian Government Research Training Program Scholarship and a Curtin University Postgraduate Scholarship top up. This research was also partly funded by the CRC for Low Carbon Living Ltd supported by the Cooperative Research Centres program, an Australian Government initiative.

**Conflicts of Interest:** The authors declare no conflict of interest.

## Nomenclature

Adapted from Cohen and Giacomo [1]:

| | |
|---|---|
| $\Delta T_{av}$ | average 30 min apparent temperature minus ambient temperature over the measurement period. |
| $Day\Delta T_{av}$ | average 30 min apparent temperature minus ambient temperature over the daytime hours of 07:00 to 19:00. |
| $\varepsilon$ | emissivity; the ratio of the energy radiated from a materials surface to that radiated from a black body or perfect emitter, ranges from 0 (perfect reflector) and 1 (perfect emitter). |
| $Eve\Delta T_{av}$ | average 30 min apparent temperature minus ambient temperature over the evening hours of 19:30 to 21:00. |
| $h_{conv}$ | the convection coefficient (W·m$^{-2}$·K$^{-1}$) |
| LE | landscape element; a common surface treatment found in a domestic garden or urban landscape. |
| $M$ | total radiant exitance; radiant power leaving (emitted, reflected and transmitted by) a surface per unit area (W·m$^{-2}$). |
| $M_e$ | emitted radiant exitance; radiant power emitted by a surface per unit area (W·m$^{-2}$). |
| $M_{ref}$ | reflected radiant exitance; radiant power reflected by a surface per unit area (W·m$^{-2}$) |
| $q_{conv}$ | the heat flux due to convection (W·m$^{-2}$) |
| SVF | sky view factor; a measure of the degree of site sky visibility [2] |
| $T_{app}$ | apparent temperature; the temperature provided by a thermal camera when the emissivity of the object is estimated (°C). |
| $T_{ref}$ | reflected temperature; the temperature of the surrounding environment (°C). |

# Appendix A

**Table A1.** App A Thermal imaging camera data (°C), Testo 876, Emissivity set to 0.95, Reflected temperature (Tref) as measured.

| Time | Tref | A | B | C | D | E | F | G | H | I | J | K | L | M | N | O | P | Q | R | S | T |
|------|------|---|---|---|---|---|---|---|---|---|---|---|---|---|---|---|---|---|---|---|---|---|
| 6:00 | −25.0 | −5.1 | −3.8 | −5.3 | −4.2 | −4.2 | −5.4 | −4.3 | −3.9 | −4.0 | −3.6 | −5.3 | −3.9 | −5.4 | −3.9 | −6.2 | −6.0 | −5.7 | −3.7 | −3.7 | −2.3 |
| 6:30 | −25.0 | −5.4 | −4.2 | −5.4 | −4.4 | −4.4 | −5.5 | −4.9 | −4.0 | −4.4 | −3.8 | −5.2 | −4.1 | −5.7 | −4.2 | −6.2 | −6.0 | −5.8 | −3.6 | −3.6 | −2.5 |
| 7:00 | −25.0 | −4.0 | −3.3 | −5.1 | −4.1 | −3.6 | −4.0 | −3.9 | −3.7 | −3.6 | −3.4 | −3.7 | −3.5 | −4.4 | −3.6 | −4.3 | −4.2 | −4.0 | −2.8 | −2.8 | −1.7 |
| 7:30 | −25.0 | −2.6 | −1.8 | −4.2 | −3.2 | −2.5 | −2.2 | −3.0 | −3.1 | −2.5 | −3.0 | −2.0 | −2.1 | −2.7 | −2.8 | −1.9 | −2.1 | −1.9 | −2.2 | −2.2 | −1.0 |
| 8:00 | −25.0 | −0.9 | −0.3 | −3.8 | −2.3 | −1.6 | −1.1 | −1.7 | −2.3 | −1.5 | −2.2 | −1.2 | −1.0 | −1.7 | −2.5 | −1.2 | −1.2 | −1.3 | −1.7 | −1.7 | −0.3 |
| 8:30 | −25.0 | 5.7 | 2.6 | −1.1 | 0.5 | 0.7 | 4.5 | 1.4 | 1.0 | 1.4 | 0.0 | 4.1 | 2.0 | 3.3 | −2.0 | 8.0 | 1.7 | −0.4 | 2.2 | 2.2 | 6.0 |
| 9:00 | −25.0 | 12.0 | 7.9 | 3.5 | 4.1 | 4.0 | 8.9 | 5.0 | 6.3 | 6.8 | 3.9 | 9.6 | 7.7 | 8.8 | 1.1 | 15.1 | 10.3 | 5.8 | 6.9 | 6.9 | 9.6 |
| 9:30 | −25.0 | 15.5 | 12.6 | 7.2 | 7.4 | 6.0 | 11.7 | 7.7 | 10.9 | 11.0 | 7.0 | 14.3 | 9.8 | 12.2 | 5.1 | 19.3 | 12.6 | 7.2 | 12.4 | 12.4 | 12.6 |
| 10:00 | −25.0 | 19.6 | 17.7 | 11.0 | 10.4 | 8.2 | 15.2 | 9.7 | 16.5 | 15.4 | 10.1 | 16.8 | 14.2 | 17.0 | 7.3 | 26.5 | 14.1 | 9.1 | 16.1 | 16.1 | 15.0 |
| 10:30 | −25.0 | 21.9 | 20.6 | 14.8 | 13.6 | 10.1 | 18.0 | 12.3 | 20.5 | 18.3 | 12.8 | 18.7 | 17.4 | 20.3 | 8.8 | 32.0 | 14.4 | 9.5 | 17.4 | 17.4 | 17.0 |
| 11:00 | −25.0 | 29.8 | 25.2 | 18.4 | 16.9 | 11.7 | 21.9 | 14.8 | 25.0 | 22.3 | 15.8 | 24.6 | 21.3 | 24.5 | 10.8 | 39.0 | 22.0 | 11.0 | 22.3 | 22.3 | 20.4 |
| 11:30 | −25.0 | 27.3 | 24.7 | 20.6 | 19.2 | 13.2 | 21.5 | 15.8 | 27.9 | 24.0 | 18.1 | 21.5 | 22.9 | 24.6 | 11.6 | 33.9 | 19.9 | 10.7 | 21.1 | 21.1 | 18.9 |
| 12:00 | −25.0 | 30.7 | 26.4 | 21.9 | 20.8 | 14.2 | 21.8 | 17.0 | 29.7 | 25.3 | 19.6 | 23.8 | 23.2 | 25.8 | 12.4 | 37.1 | 20.6 | 11.2 | 23.1 | 23.1 | 19.6 |
| 12:30 | −25.0 | 33.3 | 30.5 | 23.9 | 22.7 | 15.3 | 25.2 | 18.4 | 32.3 | 27.8 | 21.3 | 26.7 | 26.3 | 29.4 | 14.1 | 40.5 | 24.7 | 11.8 | 26.4 | 26.4 | 21.4 |
| 13:00 | −25.0 | 32.2 | 29.4 | 25.0 | 24.3 | 15.7 | 24.5 | 19.1 | 33.8 | 28.7 | 22.3 | 25.2 | 27.1 | 29.1 | 14.4 | 39.1 | 23.7 | 12.1 | 26.2 | 26.2 | 21.9 |
| 13:30 | −25.0 | 32.2 | 29.8 | 25.6 | 24.3 | 16.2 | 23.5 | 19.3 | 34.7 | 28.9 | 23.0 | 24.1 | 28.7 | 28.0 | 14.0 | 38.3 | 23.3 | 11.8 | 24.8 | 24.8 | 20.6 |
| 14:00 | −25.0 | 34.0 | 30.8 | 25.7 | 25.0 | 15.9 | 24.6 | 20.1 | 34.3 | 28.7 | 22.9 | 26.7 | 27.8 | 29.1 | 14.8 | 39.6 | 22.4 | 11.8 | 28.0 | 28.0 | 21.4 |
| 14:30 | −25.0 | 31.4 | 29.6 | 25.1 | 24.1 | 15.6 | 23.4 | 20.0 | 33.4 | 27.4 | 22.2 | 25.4 | 28.3 | 27.3 | 14.3 | 33.9 | 19.5 | 10.9 | 26.3 | 26.3 | 20.3 |
| 15:00 | −25.0 | 26.4 | 27.8 | 23.8 | 23.1 | 15.1 | 21.3 | 18.8 | 31.9 | 26.1 | 21.8 | 23.7 | 26.8 | 24.9 | 13.7 | 30.5 | 17.1 | 11.0 | 24.0 | 24.0 | 18.8 |
| 15:30 | −25.0 | 24.6 | 26.6 | 22.7 | 22.5 | 13.9 | 19.1 | 18.3 | 29.8 | 24.4 | 20.5 | 21.8 | 25.9 | 23.1 | 12.5 | 30.2 | 15.2 | 9.2 | 22.9 | 22.9 | 18.0 |
| 16:00 | −25.0 | 18.7 | 23.7 | 18.8 | 19.7 | 12.0 | 14.9 | 15.8 | 26.1 | 20.2 | 18.5 | 18.2 | 22.0 | 18.4 | 10.3 | 20.4 | 10.7 | 7.7 | 17.1 | 17.1 | 11.7 |
| 16:30 | −25.0 | 8.7 | 17.0 | 15.0 | 16.7 | 10.1 | 9.1 | 9.9 | 20.6 | 14.9 | 15.2 | 11.8 | 15.9 | 11.9 | 8.0 | 7.6 | 4.9 | 4.1 | 10.9 | 10.9 | 8.8 |
| 17:00 | −25.0 | 6.6 | 15.0 | 12.1 | 14.3 | 8.3 | 6.6 | 11.2 | 16.6 | 11.9 | 12.2 | 8.7 | 13.8 | 9.6 | 6.1 | 6.2 | 2.8 | 2.6 | 9.1 | 9.1 | 6.9 |
| 17:30 | −25.0 | 2.2 | 11.7 | 9.1 | 12.2 | 6.1 | 3.3 | 8.9 | 12.7 | 8.4 | 9.5 | 6.6 | 10.2 | 6.3 | 4.1 | 1.8 | 0.1 | −0.2 | 6.4 | 6.4 | 4.6 |
| 18:00 | −25.0 | 0.4 | 9.5 | 7.0 | 9.9 | 4.7 | 2.4 | 7.2 | 10.1 | 6.4 | 8.0 | 4.7 | 7.8 | 4.5 | 2.5 | 0.3 | −1.0 | −0.3 | 5.2 | 5.2 | 3.7 |
| 18:30 | −25.0 | −2.4 | 5.1 | 2.1 | 6.0 | 0.4 | −1.4 | 3.2 | 5.0 | 1.6 | 3.5 | 0.9 | 2.7 | 0.6 | −1.1 | −2.1 | −3.3 | −3.1 | 1.3 | 1.3 | 0.1 |
| 19:00 | −25.0 | −1.1 | 6.2 | 3.7 | 7.3 | 2.5 | 0.0 | 4.6 | 6.3 | 3.3 | 5.2 | 2.1 | 4.7 | 1.9 | 0.8 | −1.3 | −1.8 | −1.4 | 2.5 | 2.5 | 1.9 |
| 19:30 | −25.0 | −1.8 | 5.1 | 2.6 | 6.4 | 1.5 | −1.2 | 3.7 | 4.8 | 2.1 | 3.7 | 0.5 | 3.2 | 0.5 | 0.0 | −2.4 | −2.4 | −2.1 | 1.6 | 1.6 | 1.8 |
| 20:00 | −25.0 | −2.3 | 3.9 | 1.1 | 4.7 | 0.1 | −2.1 | 2.7 | 3.4 | 1.1 | 2.8 | −0.1 | 1.5 | −0.6 | −0.8 | −2.8 | −2.7 | −2.1 | 0.5 | 0.5 | 0.9 |

**Table A1.** *Cont.*

**29/09/2016, spring**

| Time | Tref | A | B | C | D | E | F | G | H | I | J | K | L | M | N | O | P | Q | R | S | T |
|------|------|-----|-----|-----|-----|-----|-----|-----|-----|-----|-----|-----|-----|-----|-----|-----|-----|-----|-----|-----|-----|
| 5:00 | −25.0 | −2.3 | −0.8 | −1.9 | −0.3 | −1.3 | −4.3 | −1.1 | −0.3 | −0.8 | 0.2 | −3.8 | −0.5 | −3.9 | −4.7 | −3.4 | −3.8 | −3.2 | −5.0 | | 2.0 |
| 5:30 | −25.0 | −2.6 | −1.6 | −2.3 | −0.8 | −2.0 | −5.2 | −2.1 | −0.8 | −1.1 | −0.3 | −4.9 | −0.9 | −5.0 | −5.0 | −3.9 | −4.2 | −3.5 | −5.8 | | 1.5 |
| 6:00 | −25.0 | −2.6 | −2.0 | −2.9 | −1.5 | −2.5 | −4.8 | −2.7 | −1.1 | −1.3 | −0.6 | −5.1 | −1.4 | −4.7 | −4.8 | −3.5 | −1.4 | −3.6 | −5.7 | | 1.2 |
| 6:30 | −25.0 | −0.3 | −1.2 | −2.6 | −1.3 | −2.3 | −3.5 | −2.0 | −0.6 | −0.8 | −0.4 | −2.8 | −0.9 | −3.7 | −4.0 | −0.8 | −1.3 | −2.6 | −3.4 | | 1.8 |
| 7:00 | −25.0 | 3.3 | 0.4 | −1.3 | −0.1 | −1.1 | 0.1 | −0.2 | 0.6 | 0.3 | 0.6 | 0.9 | 0.7 | −1.0 | −3.3 | 2.9 | −0.6 | 0.1 | 0.7 | | 4.8 |
| 7:30 | −25.0 | 14.1 | 9.2 | 3.8 | 4.8 | 3.8 | 9.0 | 4.3 | 7.0 | 5.6 | 4.3 | 11.5 | 7.0 | 9.2 | 3.0 | 15.9 | 11.2 | 7.1 | 13.7 | | 10.8 |
| 8:00 | −25.0 | 16.9 | 15.3 | 8.6 | 8.9 | 6.4 | 13.1 | 7.6 | 13.4 | 10.3 | 9.0 | 17.3 | 12.8 | 15.6 | 6.0 | 19.1 | 13.5 | 8.7 | 19.8 | | 13.6 |
| 8:30 | −25.0 | 24.6 | 21.4 | 13.6 | 12.9 | 9.3 | 18.1 | 9.9 | 19.5 | 14.8 | 12.9 | 21.8 | 17.7 | 22.4 | 9.1 | 30.3 | 15.4 | 10.9 | 28.2 | | 16.0 |
| 9:00 | −25.0 | 32.4 | 27.6 | 18.2 | 17.0 | 12.2 | 22.7 | 12.5 | 26.0 | 19.6 | 17.4 | 26.8 | 23.3 | 27.0 | 12.5 | 40.4 | 23.7 | 12.8 | 33.9 | | 19.0 |
| 9:30 | −25.0 | 40.1 | 33.6 | 22.3 | 21.1 | 15.0 | 26.7 | 14.9 | 31.0 | 23.6 | 20.3 | 32.6 | 27.8 | 33.3 | 15.4 | 48.2 | 29.9 | 13.5 | 42.4 | | 22.3 |
| 10:00 | −25.0 | 45.0 | 38.5 | 26.9 | 25.4 | 16.6 | 30.0 | 17.4 | 35.8 | 26.8 | 24.2 | 37.1 | 31.1 | 38.8 | 18.9 | 52.1 | 32.7 | 14.1 | 46.6 | | 23.4 |
| 10:30 | −25.0 | 44.9 | 41.5 | 30.0 | 28.4 | 18.3 | 32.0 | 19.3 | 39.6 | 29.2 | 26.9 | 36.4 | 34.4 | 41.3 | 20.9 | 51.3 | 32.6 | 15.0 | 48.8 | | 26.2 |
| 11:00 | −25.0 | 47.7 | 45.7 | 33.2 | 31.7 | 20.0 | 35.4 | 21.6 | 43.4 | 32.1 | 29.7 | 38.1 | 36.5 | 44.4 | 23.4 | 55.4 | 36.7 | 16.2 | 54.1 | | 27.8 |
| 11:30 | −25.0 | 55.5 | 52.0 | 36.4 | 34.7 | 22.1 | 38.9 | 23.4 | 48.0 | 35.5 | 32.6 | 46.7 | 40.2 | 49.4 | 25.9 | 63.7 | 42.3 | 17.8 | 60.0 | | 29.8 |
| 12:00 | −25.0 | 52.3 | 51.7 | 37.9 | 36.1 | 22.3 | 38.1 | 24.1 | 49.1 | 35.8 | 33.7 | 45.1 | 41.6 | 48.0 | 26.0 | 60.3 | 39.2 | 18.2 | 55.9 | | 30.1 |
| 12:30 | −25.0 | 54.7 | 53.7 | 39.0 | 37.4 | 23.2 | 40.0 | 24.5 | 50.7 | 36.8 | 35.1 | 48.8 | 42.3 | 50.2 | 27.7 | 61.2 | 40.7 | 19.5 | 59.5 | | 30.4 |
| 13:00 | −25.0 | 49.4 | 52.4 | 38.9 | 37.7 | 23.4 | 37.5 | 24.7 | 50.4 | 36.3 | 35.3 | 45.7 | 42.7 | 49.1 | 26.7 | 58.3 | 38.3 | 19.3 | 56.2 | | 30.2 |
| 13:30 | −25.0 | 52.2 | 54.9 | 39.7 | 38.7 | 23.6 | 39.2 | 25.3 | 50.7 | 37.3 | 35.6 | 47.8 | 42.6 | 50.1 | 27.2 | 59.6 | 38.9 | 19.3 | 56.6 | | 30.1 |
| 14:00 | −25.0 | 46.4 | 53.1 | 38.9 | 37.9 | 23.5 | 36.5 | 25.1 | 50.2 | 36.9 | 35.4 | 46.4 | 41.8 | 47.0 | 26.2 | 53.4 | 35.6 | 18.8 | 51.4 | | 29.4 |
| 14:30 | −25.0 | 42.6 | 50.7 | 37.1 | 36.3 | 21.5 | 33.3 | 24.3 | 47.8 | 34.7 | 34.3 | 41.0 | 36.9 | 43.4 | 24.7 | 48.9 | 32.0 | 17.1 | 47.4 | | 27.9 |
| 15:00 | −25.0 | 42.0 | 49.3 | 36.0 | 35.4 | 20.8 | 33.1 | 23.6 | 45.6 | 33.7 | 33.2 | 40.1 | 34.7 | 42.7 | 23.4 | 50.0 | 30.4 | 16.5 | 49.1 | | 28.6 |
| 15:30 | −25.0 | 35.0 | 46.7 | 33.9 | 33.5 | 20.4 | 29.9 | 22.3 | 43.1 | 32.1 | 31.5 | 37.3 | 34.0 | 38.8 | 21.8 | 43.6 | 26.5 | 15.7 | 43.2 | | 25.2 |
| 16:00 | −25.0 | 27.2 | 40.0 | 30.4 | 30.5 | 17.9 | 25.0 | 20.5 | 39.3 | 28.5 | 28.6 | 30.3 | 28.5 | 32.9 | 18.9 | 32.5 | 20.6 | 13.8 | 34.9 | | 22.7 |
| 16:30 | −25.0 | 24.7 | 36.6 | 27.6 | 28.1 | 16.6 | 22.1 | 18.9 | 35.4 | 26.0 | 26.5 | 25.9 | 26.5 | 28.5 | 16.9 | 30.4 | 17.3 | 12.4 | 30.8 | | 20.5 |
| 17:00 | −25.0 | 18.3 | 30.7 | 23.3 | 24.6 | 14.3 | 17.5 | 16.3 | 29.1 | 20.4 | 22.4 | 21.2 | 20.9 | 22.9 | 12.9 | 22.3 | 12.5 | 9.3 | 24.0 | | 17.2 |
| 17:30 | −25.0 | 11.5 | 21.8 | 18.0 | 20.3 | 11.8 | 10.5 | 13.6 | 22.4 | 16.0 | 18.5 | 14.3 | 15.7 | 15.0 | 9.7 | 10.3 | 7.1 | 7.2 | 14.6 | | 13.1 |
| 18:00 | −25.0 | 9.9 | 17.6 | 14.5 | 17.1 | 10.5 | 8.5 | 11.6 | 18.1 | 12.8 | 15.3 | 12.0 | 12.6 | 11.9 | 7.8 | 8.6 | 5.6 | 6.3 | 12.2 | | 12.0 |
| 18:30 | −25.0 | 8.0 | 14.3 | 11.7 | 14.9 | −1.3 | 6.9 | 10.3 | 15.0 | 11.0 | 13.2 | 9.7 | 10.9 | 9.2 | 6.5 | 6.5 | 4.5 | | 9.6 | | 11.0 |
| 19:00 | −25.0 | 6.9 | 12.2 | 9.8 | 13.4 | −2.0 | 5.3 | 9.1 | 12.8 | 9.8 | 11.5 | 8.3 | 9.7 | 7.9 | 5.1 | 5.5 | 4.0 | 4.7 | 7.7 | | 10.0 |
| 19:30 | −25.0 | 4.6 | 10.1 | 8.1 | 12.0 | −2.5 | 3.4 | 8.1 | 11.2 | 8.4 | 10.1 | 5.0 | 8.4 | 5.7 | 3.6 | 3.7 | 2.6 | 3.4 | 4.6 | | 8.9 |
| 20:00 | −25.0 | 4.5 | 8.4 | 7.1 | 10.5 | −2.3 | 2.4 | 6.5 | 9.7 | 7.6 | 8.9 | 3.5 | 7.5 | 4.0 | 2.2 | 3.3 | 2.2 | 2.9 | 3.3 | | 8.1 |
| 20:30 | −25.0 | 4.5 | 7.1 | 6.0 | 9.6 | −1.1 | 2.3 | 5.9 | 8.5 | 6.5 | 8.0 | 3.9 | 6.4 | 3.6 | 1.6 | 3.4 | 2.2 | 3.4 | 3.1 | | 8.0 |

**Table A1.** *Cont.*

### 09/01/2018, summer

| Time | Tref | A | B | C | D | E | F | G | H | I | J | K | L | M | N | O | P | Q | R | S | T |
|------|------|---|---|---|---|---|---|---|---|---|---|---|---|---|---|---|---|---|---|---|---|---|
| 5:00 | −15.5 | 12.0 | 13.3 | 12.3 | 14.1 | 13.5 | 12.3 | 13.7 | 13.2 | 12.8 | 12.7 | 14.4 | 14.5 | 11.9 | 11.2 | 12.0 | 11.1 | 11.1 | 13.5 | 12.0 | 13.5 |
| 5:30 | −14.7 | 12.4 | 13.4 | 12.3 | 14.0 | 13.4 | 12.4 | 13.5 | 13.1 | 12.7 | 12.6 | 14.7 | 14.3 | 12.1 | 11.1 | 12.5 | 11.2 | 10.7 | 13.5 | 12.3 | 13.6 |
| 6:00 | −14.0 | 13.6 | 13.8 | 12.6 | 14.1 | 13.2 | 13.3 | 13.8 | 13.8 | 13.0 | 13.0 | 15.5 | 14.5 | 13.0 | 11.6 | 14.5 | 11.9 | 11.6 | 14.9 | 13.0 | 14.6 |
| 6:30 | −13.2 | 18.2 | 17.5 | 14.5 | 15.6 | 14.7 | 16.9 | 15.4 | 16.0 | 15.8 | 14.6 | 19.1 | 17.2 | 16.7 | 13.6 | 19.7 | 15.1 | 14.2 | 20.3 | 15.9 | 17.7 |
| 7:00 | −13.7 | 23.6 | 22.4 | 17.7 | 18.5 | 16.7 | 21.1 | 17.3 | 20.4 | 20.0 | 17.2 | 23.9 | 21.3 | 22.1 | 16.1 | 29.0 | 19.8 | 16.8 | 26.0 | 18.8 | 20.3 |
| 7:30 | −11.0 | 31.1 | 28.2 | 21.6 | 21.2 | 19.0 | 25.7 | 19.3 | 25.6 | 24.7 | 20.9 | 28.5 | 25.9 | 27.2 | 19.0 | 36.4 | 25.4 | 19.7 | 32.3 | 22.9 | 23.9 |
| 8:00 | −11.5 | 36.9 | 34.1 | 26.4 | 25.5 | 21.6 | 30.9 | 22.0 | 31.3 | 29.7 | 25.2 | 33.7 | 30.6 | 33.0 | 21.9 | 42.6 | 29.7 | 22.4 | 38.4 | 27.2 | 28.5 |
| 8:30 | −9.8 | 45.1 | 40.7 | 31.1 | 29.3 | 24.5 | 35.1 | 24.7 | 37.8 | 35.2 | 29.8 | 39.2 | 35.9 | 39.1 | 24.9 | 50.5 | 36.1 | 25.8 | 45.0 | 32.2 | 31.1 |
| 9:00 | −9.9 | 51.1 | 46.1 | 35.9 | 33.3 | 27.5 | 39.3 | 27.5 | 43.7 | 40.2 | 34.3 | 44.1 | 40.2 | 44.3 | 28.2 | 54.7 | 38.7 | 27.7 | 52.5 | 38.2 | 34.1 |
| 9:30 | −9.2 | 56.2 | 51.6 | 39.9 | 37.6 | 30.3 | 42.4 | 30.6 | 49.5 | 45.1 | 38.5 | 49.1 | 44.7 | 49.2 | 31.2 | 63.1 | 44.9 | 31.0 | 56.0 | 40.2 | 36.4 |
| 10:00 | −8.4 | 62.0 | 56.1 | 43.6 | 41.2 | 33.1 | 46.7 | 33.3 | 54.8 | 50.1 | 42.3 | 50.2 | 49.4 | 53.3 | 34.1 | 67.3 | 48.3 | 32.5 | 61.3 | 46.7 | 38.8 |
| 10:30 | −7.8 | 64.8 | 61.1 | 47.7 | 44.9 | 35.0 | 50.8 | 34.9 | 59.4 | 53.8 | 46.1 | 57.8 | 52.8 | 58.5 | 37.1 | 70.9 | 51.5 | 34.8 | 66.8 | 53.1 | 41.0 |
| 11:00 | −6.1 | 70.4 | 65.9 | 51.4 | 48.4 | 37.5 | 54.6 | 37.1 | 63.7 | 57.6 | 49.6 | 62.4 | 57.0 | 62.6 | 39.1 | 75.9 | 54.6 | 36.0 | 71.5 | 57.8 | 43.9 |
| 11:30 | −5.6 | 75.0 | 69.0 | 53.9 | 51.1 | 39.6 | 55.5 | 38.7 | 67.4 | 61.2 | 52.7 | 65.0 | 60.8 | 65.2 | 41.2 | 79.9 | 57.9 | 38.1 | 73.5 | 60.6 | 43.5 |
| 12:00 | −5.4 | 72.0 | 71.0 | 55.9 | 53.6 | 41.0 | 58.3 | 40.4 | 69.4 | 62.5 | 54.3 | 65.1 | 62.7 | 67.3 | 42.7 | 78.3 | 57.3 | 38.0 | 77.4 | 64.7 | 46.2 |
| 12:30 | −4.8 | 74.6 | 73.9 | 57.4 | 55.2 | 43.2 | 58.1 | 41.5 | 71.8 | 65.1 | 56.7 | 70.1 | 66.1 | 69.5 | 44.3 | 81.4 | 59.4 | 40.3 | 78.5 | 65.7 | 44.7 |
| 13:00 | −3.8 | 73.4 | 72.7 | 57.6 | 56.0 | 43.4 | 58.0 | 41.8 | 72.5 | 65.7 | 57.5 | 64.2 | 66.6 | 67.5 | 44.3 | 79.0 | 57.5 | 39.2 | 76.4 | 66.2 | 44.8 |
| 13:30 | −2.1 | 69.2 | 72.7 | 57.6 | 56.3 | 44.3 | 54.2 | 41.6 | 73.3 | 65.9 | 58.0 | 64.7 | 67.2 | 67.5 | 44.2 | 76.1 | 56.8 | 39.5 | 71.0 | 59.6 | 40.9 |
| 14:00 | −2.8 | 68.1 | 70.2 | 55.6 | 55.3 | 42.3 | 53.2 | 41.4 | 70.6 | 63.0 | 56.8 | 62.7 | 63.6 | 65.3 | 43.1 | 74.7 | 53.8 | 38.2 | 68.2 | 61.3 | 42.8 |
| 14:30 | −2.6 | 58.1 | 65.1 | 53.5 | 53.5 | 40.8 | 49.5 | 39.7 | 68.3 | 59.8 | 55.0 | 54.0 | 60.8 | 59.0 | 40.6 | 63.9 | 48.2 | 35.8 | 63.0 | 60.1 | 40.6 |
| 15:00 | −3.4 | 55.4 | 63.7 | 51.6 | 52.1 | 39.4 | 48.6 | 38.7 | 66.3 | 57.5 | 53.3 | 53.8 | 59.0 | 58.2 | 39.5 | 63.8 | 46.9 | 34.6 | 62.1 | 57.0 | 39.7 |
| 15:30 | −4.1 | 52.4 | 60.9 | 50.0 | 50.6 | 38.3 | 46.2 | 38.0 | 64.4 | 56.1 | 51.7 | 51.9 | 56.8 | 54.5 | 38.2 | 61.0 | 45.2 | 33.8 | 57.2 | 53.9 | 39.7 |
| 16:00 | −3.5 | 48.5 | 56.7 | 46.9 | 47.6 | 36.9 | 42.0 | 36.5 | 60.8 | 52.5 | 49.2 | 48.1 | 54.1 | 50.3 | 35.7 | 54.3 | 40.5 | 31.7 | 52.5 | 49.3 | 35.9 |
| 16:30 | −5.6 | 43.0 | 51.7 | 44.3 | 45.4 | 34.1 | 39.0 | 34.6 | 56.6 | 47.9 | 46.0 | 42.8 | 49.9 | 45.7 | 33.1 | 49.7 | 35.9 | 28.6 | 48.2 | 46.7 | 35.2 |
| 17:00 | −5.0 | 37.2 | 47.4 | 40.9 | 42.3 | 31.8 | 35.6 | 32.5 | 52.7 | 43.9 | 43.2 | 39.7 | 45.8 | 41.2 | 30.8 | 42.8 | 32.4 | 27.1 | 43.0 | 41.0 | 32.5 |
| 17:30 | −5.1 | 34.5 | 43.5 | 37.3 | 39.0 | 29.7 | 31.7 | 30.5 | 47.9 | 40.1 | 39.9 | 36.5 | 41.8 | 37.4 | 28.6 | 39.2 | 29.4 | 25.1 | 38.6 | 36.9 | 29.7 |
| 18:00 | −6.6 | 29.2 | 39.2 | 34.0 | 36.2 | 27.9 | 29.1 | 28.9 | 43.2 | 35.9 | 36.2 | 32.5 | 38.1 | 33.1 | 26.3 | 32.8 | 25.6 | 22.8 | 33.9 | 33.1 | 27.1 |
| 18:30 | −6.1 | 24.1 | 33.6 | 29.7 | 32.6 | 25.0 | 24.8 | 26.4 | 37.5 | 30.2 | 32.2 | 27.8 | 32.9 | 28.3 | 23.3 | 25.7 | 21.7 | 19.9 | 28.4 | 27.4 | 24.0 |
| 19:00 | −6.6 | 20.8 | 29.1 | 26.7 | 29.8 | 23.4 | 21.8 | 24.6 | 32.9 | 27.1 | 28.9 | 24.3 | 29.7 | 24.6 | 21.2 | 21.2 | 19.0 | 18.2 | 24.6 | 23.8 | 21.7 |
| 19:30 | −6.8 | 19.1 | 25.9 | 23.8 | 27.6 | 21.9 | 19.6 | 23.4 | 28.6 | 24.2 | 25.8 | 22.5 | 26.5 | 21.7 | 19.3 | 18.9 | 17.7 | 17.1 | 22.5 | 21.8 | 20.4 |
| 20:00 | −4.1 | 19.2 | 23.9 | 22.5 | 25.7 | 20.8 | 19.7 | 22.4 | 26.1 | 22.3 | 24.0 | 22.3 | 24.2 | 20.7 | 19.1 | 18.5 | 17.6 | 17.6 | 22.0 | 21.1 | 20.5 |
| 20:30 | −4.3 | 18.9 | 22.3 | 21.1 | 24.4 | 20.3 | 19.0 | 21.7 | 23.7 | 21.4 | 22.5 | 21.3 | 23.4 | 19.9 | 18.3 | 18.1 | 17.1 | 17.2 | 20.9 | 20.2 | 19.5 |
| 21:00 | −3.0 | 18.4 | 21.4 | 20.1 | 23.5 | 19.6 | 18.7 | 21.2 | 22.7 | 20.5 | 21.2 | 20.6 | 22.4 | 18.9 | 17.6 | 17.9 | 17.1 | 16.7 | 19.7 | 19.8 | 19.4 |
| 21:30 | −2.3 | 18.6 | 20.5 | 19.6 | 22.5 | 19.4 | 19.0 | 21.0 | 21.7 | 20.1 | 20.5 | 20.8 | 21.9 | 18.9 | 17.6 | 18.3 | 17.3 | 16.9 | 20.1 | 19.7 | 19.1 |

**Table A1.** *Cont.*

**09/04/2017, autumn**

| Time | Tref | A | B | C | D | E | F | G | H | I | J | K | L | M | N | O | P | Q | R | S | T |
|------|------|---|---|---|---|---|---|---|---|---|---|---|---|---|---|---|---|---|---|---|---|
| 5:00 | −20.2 | 6.0 | 5.3 | 3.4 | 5.5 | 4.7 | 4.9 | 5.6 | 4.6 | 5.1 | 4.6 | 6.3 | 5.8 | 4.4 | 4.5 | 4.8 | 4.7 | 5.3 | 4.6 | 5.3 | 5.5 |
| 5:30 | −20.1 | 3.9 | 4.8 | 2.8 | 4.9 | | 4.2 | 5.3 | 4.6 | 4.3 | 4.4 | 4.2 | 5.5 | 3.2 | 2.6 | 3.1 | 3.5 | 3.8 | 3.2 | 4.6 | 4.2 |
| 6:00 | −20.4 | 8.0 | 6.4 | 3.5 | 5.8 | 6.4 | 7.2 | 6.6 | 5.4 | 5.9 | 5.1 | 8.4 | 7.2 | 5.4 | 4.6 | 6.8 | 6.7 | 7.2 | 6.7 | 6.5 | 7.7 |
| 6:30 | −20.7 | 5.8 | 5.9 | 3.9 | 5.7 | 6.5 | 6.2 | 6.4 | 5.4 | 5.5 | 5.0 | 7.0 | 7.3 | 5.0 | 4.0 | 5.1 | 5.1 | 5.4 | 5.2 | 6.1 | 5.9 |
| 7:00 | −20.4 | 8.0 | 7.2 | 4.1 | 5.9 | 6.3 | 7.2 | 7.0 | 5.9 | 6.3 | 5.5 | 8.3 | 7.7 | 5.8 | 4.7 | 7.7 | 6.4 | 6.3 | 7.2 | 7.2 | 7.4 |
| 7:30 | −20.4 | 10.0 | 7.9 | 5.2 | 6.6 | 7.3 | 8.5 | 7.6 | 6.8 | 7.3 | 6.3 | 9.5 | 8.6 | 6.5 | 5.0 | 9.3 | 8.3 | 7.4 | 8.5 | 8.5 | 9.4 |
| 8:00 | −19.8 | 17.1 | 13.8 | 9.1 | 9.8 | 9.6 | 15.3 | 11.4 | 11.6 | 11.7 | 9.6 | 17.0 | 12.9 | 13.8 | 8.9 | 18.8 | 15.1 | 12.3 | 17.5 | 15.6 | 16.1 |
| 8:30 | −19.0 | 22.5 | 19.3 | 14.3 | 13.7 | 12.7 | 21.0 | 14.4 | 17.9 | 16.5 | 13.7 | 23.4 | 19.8 | 19.7 | 11.3 | 24.0 | 19.5 | 15.8 | 24.9 | 20.2 | 21.1 |
| 9:00 | −18.3 | 29.5 | 26.9 | 18.6 | 16.9 | 14.9 | 25.5 | 17.5 | 24.8 | 22.1 | 17.7 | 26.9 | 25.7 | 25.6 | 13.9 | 35.2 | 22.2 | 17.5 | 31.8 | 23.8 | 23.4 |
| 9:30 | −18.7 | 35.8 | 33.5 | 24.1 | 21.2 | 17.4 | 29.6 | 20.9 | 31.1 | 27.3 | 22.1 | 31.4 | 29.6 | 31.2 | 17.2 | 45.6 | 25.1 | 18.9 | 37.6 | 27.6 | 28.0 |
| 10:00 | −17.5 | 45.0 | 40.1 | 28.5 | 24.9 | 20.2 | 33.4 | 23.0 | 36.7 | 31.9 | 26.1 | 35.4 | 34.8 | 36.8 | 20.6 | 53.0 | 34.7 | 19.6 | 45.5 | 31.3 | 29.5 |
| 10:30 | −17.4 | 49.7 | 45.4 | 32.9 | 28.9 | 22.8 | 36.3 | 26.1 | 42.0 | 35.9 | 30.1 | 39.9 | 38.6 | 40.4 | 22.9 | 56.7 | 39.1 | 21.2 | 48.6 | 35.2 | 33.7 |
| 11:00 | −17.0 | 50.3 | 48.8 | 36.5 | 32.1 | 25.2 | 37.3 | 28.0 | 46.0 | 39.8 | 33.3 | 39.2 | 41.8 | 43.8 | 25.0 | 58.8 | 40.5 | 23.2 | 51.3 | 39.8 | 35.0 |
| 11:30 | −15.6 | 53.3 | 53.2 | 39.7 | 35.2 | 26.7 | 40.1 | 30.3 | 49.1 | 42.5 | 36.7 | 42.5 | 43.7 | 47.3 | 27.5 | 62.1 | 43.0 | 26.7 | 55.1 | 42.8 | 37.3 |
| 12:00 | −15.1 | 56.2 | 55.6 | 42.3 | 37.7 | 28.4 | 41.1 | 31.7 | 51.8 | 45.0 | 38.8 | 45.8 | 47.7 | 50.2 | 29.4 | 63.9 | 45.1 | 28.4 | 58.3 | 45.2 | 37.8 |
| 12:30 | −15.9 | 55.5 | 57.4 | 43.7 | 39.8 | 29.5 | 41.1 | 32.0 | 54.0 | 46.8 | 40.7 | 47.3 | 49.5 | 51.2 | 29.9 | 64.2 | 45.5 | 28.6 | 58.7 | 43.6 | 36.1 |
| 13:00 | −14.5 | 58.3 | 59.1 | 45.2 | 41.7 | 29.8 | 42.9 | 33.3 | 54.2 | 48.1 | 41.8 | 48.0 | 49.9 | 52.0 | 31.3 | 65.5 | 44.9 | 28.7 | 58.4 | 48.8 | 38.8 |
| 13:30 | −15.8 | 51.1 | 59.0 | 45.6 | 42.3 | 30.3 | 42.6 | 33.9 | 53.9 | 47.5 | 42.3 | 46.2 | 50.6 | 52.0 | 31.4 | 62.2 | 42.8 | 28.1 | 59.9 | 48.5 | 38.7 |
| 14:00 | −15.1 | 53.8 | 58.1 | 45.1 | 43.3 | 30.2 | 41.4 | 33.9 | 53.6 | 47.8 | 42.3 | 48.7 | 50.1 | 51.7 | 31.4 | 59.6 | 42.0 | 27.5 | 59.8 | 49.8 | 37.2 |
| 14:30 | −14.3 | 51.2 | 57.0 | 43.9 | 42.5 | 29.7 | 38.6 | 32.9 | 52.0 | 46.5 | 41.7 | 46.0 | 49.5 | 49.2 | 30.5 | 57.8 | 39.3 | 25.6 | 54.0 | 45.8 | 34.7 |
| 15:00 | −13.9 | 47.6 | 53.6 | 42.2 | 41.1 | 29.0 | 37.2 | 31.9 | 49.7 | 44.0 | 40.5 | 43.9 | 48.1 | 46.5 | 28.9 | 53.6 | 36.2 | 24.9 | 50.7 | 43.8 | 33.6 |
| 15:30 | −12.9 | 40.7 | 50.8 | 40.1 | 39.5 | 27.6 | 33.7 | 30.6 | 46.9 | 41.4 | 38.1 | 39.7 | 45.9 | 43.0 | 27.4 | 47.3 | 32.5 | 22.2 | 46.3 | 39.8 | 30.5 |
| 16:00 | −14.6 | 33.1 | 44.1 | 36.5 | 36.8 | 25.3 | 29.0 | 28.1 | 42.8 | 36.8 | 35.5 | 33.3 | 39.9 | 36.7 | 24.6 | 39.5 | 27.2 | 20.6 | 37.4 | 35.0 | 27.4 |
| 16:30 | −15.5 | 27.1 | 38.2 | 32.0 | 33.3 | 23.3 | 24.0 | 24.9 | 37.2 | 32.3 | 31.8 | 28.8 | 34.7 | 30.9 | 21.5 | 30.3 | 22.1 | 17.8 | 31.8 | 28.9 | 23.0 |
| 17:00 | −16.4 | 22.5 | 33.2 | 27.8 | 30.2 | 20.7 | 21.1 | 22.5 | 32.4 | 27.8 | 28.2 | 25.1 | 31.1 | 26.4 | 19.2 | 25.2 | 18.4 | 15.3 | 27.0 | 24.7 | 20.4 |
| 17:30 | −18.1 | 15.9 | 28.2 | 23.8 | 26.6 | 18.6 | 17.4 | 19.7 | 27.4 | 23.1 | 24.2 | 20.4 | 26.4 | 21.6 | 16.9 | 16.1 | 13.8 | 12.6 | 21.8 | 20.2 | 16.3 |
| 18:00 | −17.6 | 12.4 | 23.6 | 20.2 | 24.1 | 16.4 | 14.1 | 17.8 | 23.2 | 19.4 | 21.4 | 16.7 | 22.4 | 18.1 | 14.6 | 11.7 | 10.9 | 10.8 | 16.7 | 16.3 | 13.7 |
| 18:30 | −18.6 | 10.7 | 20.2 | 17.4 | 21.8 | 14.6 | 11.5 | 16.0 | 19.9 | 16.8 | 18.3 | 13.3 | 19.7 | 15.3 | 12.5 | 9.8 | 9.4 | 9.7 | 13.6 | 14.3 | 11.0 |
| 19:00 | −19.1 | 8.9 | 17.5 | 15.1 | 19.9 | 13.0 | 9.6 | 14.3 | 17.1 | 14.7 | 16.0 | 10.7 | 17.7 | 13.2 | 11.1 | 7.9 | 7.7 | 7.5 | 11.1 | 12.6 | 9.0 |
| 19:30 | −18.6 | 8.3 | 15.1 | 12.3 | 18.0 | 11.4 | 7.4 | 12.5 | 15.4 | 12.8 | 14.2 | 9.2 | 15.4 | 11.3 | 9.5 | 6.7 | 6.9 | 7.0 | 9.1 | 10.8 | 8.4 |
| 20:00 | −19.4 | 8.5 | 13.5 | 11.3 | 16.5 | 10.2 | 7.8 | 11.4 | 13.7 | 11.8 | 12.6 | 9.3 | 14.0 | 10.1 | 8.4 | 6.8 | 6.4 | 6.7 | 9.2 | 10.8 | 8.1 |
| 20:30 | −19.5 | 7.2 | 11.6 | 9.9 | 14.9 | 9.3 | 6.7 | 10.2 | 12.0 | 10.2 | 11.0 | 7.1 | 12.2 | 8.7 | 7.5 | 5.7 | 5.9 | 6.1 | 7.4 | 9.3 | 6.8 |
| 21:00 | −19.7 | 7.7 | 10.4 | 8.5 | 13.5 | 8.5 | 7.1 | 10.0 | 10.8 | 9.2 | 10.2 | 8.9 | 11.1 | 8.3 | 6.9 | 6.5 | 6.2 | 6.7 | 8.3 | 9.1 | 8.1 |

**Table A1.** *Cont.*

| Time | Tref | A | B | C | E | F | H | I | J | K | R | S | T | U | V |
|------|------|---|---|---|---|---|---|---|---|---|---|---|---|---|---|
| **21/01/2019, summer** | | | | | | | | | | | | | | | |
| 5:00 | −7.6 | 16.3 | 17.3 | 16.9 | 17.0 | 14.2 | 17.9 | 17.7 | 17.8 | 15.6 | 15.0 | 15.4 | 15.5 | 5:01 | 17.3 |
| 5:30 | −7.7 | 16.0 | 17.0 | 16.6 | 16.8 | 14.3 | 17.4 | 17.3 | 17.4 | 15.1 | 14.7 | 20.4 | 15.6 | 0:39 | 17.8 |
| 6:00 | −7.4 | 17.5 | 17.7 | 16.8 | 17.2 | 15.7 | 17.8 | 17.9 | 17.6 | 16.8 | 16.4 | 18.8 | 17.2 | 14:42 | 18.3 |
| 6:30 | −6.5 | 19.4 | 19.3 | 17.5 | 18.1 | 18.2 | 19.5 | 19.0 | 18.2 | 19.3 | 19.4 | 19.6 | 20.9 | 11:49 | 20.3 |
| 7:00 | −0.1 | 24.9 | 25.7 | 20.0 | 20.0 | 23.4 | 22.9 | 23.1 | 20.6 | 25.3 | 24.8 | 24.2 | 24.0 | 15:45 | 22.3 |
| 7:30 | −3.5 | 33.2 | 31.6 | 22.5 | 21.7 | 27.3 | 26.7 | 26.9 | 23.4 | 30.6 | 30.2 | 27.9 | 28.0 | 2:33 | 23.6 |
| 8:00 | −4.2 | 39.3 | 36.1 | 25.0 | 23.4 | 32.1 | 31.3 | 30.5 | 25.9 | 34.5 | 34.6 | 30.6 | 31.0 | 20:22 | 25.8 |
| 8:30 | −4.6 | 46.9 | 42.4 | 28.7 | 26.1 | 38.0 | 36.1 | 35.2 | 29.5 | 40.1 | 41.5 | 33.8 | 34.6 | 23:21 | 27.3 |
| 9:00 | −2.8 | 53.7 | 48.1 | 31.7 | 28.4 | 42.6 | 41.4 | 39.5 | 33.1 | 45.7 | 47.4 | 35.9 | 39.2 | 14:52 | 28.7 |
| 9:30 | −2.8 | 56.2 | 51.6 | 35.2 | 30.2 | 45.0 | 46.3 | 42.9 | 36.2 | 47.5 | 50.5 | 35.2 | 40.3 | 17:17 | 29.9 |
| 10:00 | −3.5 | 60.9 | 55.1 | 39.0 | 32.2 | 49.0 | 50.8 | 46.5 | 39.4 | 50.7 | 55.1 | 37.3 | 42.7 | 22:58 | 31.3 |
| 10:30 | −2.5 | 66.1 | 59.3 | 41.9 | 33.6 | 52.1 | 55.2 | 49.7 | 42.4 | 53.2 | 58.7 | 38.5 | 43.2 | 22:16 | 32.2 |
| 11:00 | −3.5 | 68.1 | 62.0 | 44.5 | 35.0 | 53.1 | 58.5 | 52.3 | 45.2 | 54.9 | 61.3 | 37.5 | 43.2 | 22:58 | 31.9 |
| 11:30 | −3.7 | 68.3 | 62.4 | 46.6 | 35.3 | 54.4 | 61.5 | 53.4 | 47.1 | 55.4 | 63.5 | 38.9 | 46.3 | 18:30 | 32.2 |
| 12:00 | −5.3 | 71.2 | 64.2 | 48.8 | 36.3 | 55.1 | 63.9 | 55.4 | 49.0 | 57.2 | 64.2 | 39.0 | 46.1 | 12:38 | 32.3 |
| 12:30 | −6.2 | 71.6 | 65.7 | 49.8 | 37.0 | 55.2 | 65.2 | 56.4 | 50.3 | 56.8 | 62.9 | 37.9 | 44.0 | 8:45 | 33.0 |
| 13:00 | −7.8 | 73.0 | 65.6 | 50.3 | 36.6 | 52.7 | 66.3 | 56.6 | 51.2 | 55.9 | 60.7 | 35.8 | 45.1 | 12:21 | 32.0 |
| 13:30 | −8.4 | 73.9 | 66.7 | 51.5 | 37.0 | 55.1 | 67.6 | 57.3 | 51.9 | 58.9 | 65.2 | 38.9 | 48.5 | 13:01 | 33.8 |
| 14:00 | −9.1 | 70.9 | 65.8 | 51.9 | 37.5 | 54.8 | 67.4 | 57.2 | 52.3 | 57.3 | 63.1 | 38.5 | 46.8 | 15:52 | 33.1 |
| 14:30 | −9.5 | 72.9 | 67.3 | 51.5 | 37.5 | 53.8 | 67.6 | 56.8 | 52.3 | 58.5 | 65.0 | 38.3 | 44.9 | 18:27 | 32.5 |
| 15:00 | −10.9 | 67.1 | 64.0 | 50.4 | 37.2 | 50.1 | 65.6 | 55.3 | 51.4 | 53.2 | 58.8 | 36.3 | 42.3 | 12:22 | 31.7 |
| 15:30 | −10.8 | 62.0 | 60.8 | 48.9 | 36.5 | 47.0 | 64.0 | 53.1 | 50.0 | 48.8 | 53.3 | 34.0 | 40.6 | 14:34 | 31.1 |
| 16:00 | −10.9 | 59.4 | 58.3 | 47.4 | 34.8 | 44.4 | 61.3 | 50.9 | 48.2 | 47.6 | 53.4 | 33.9 | 39.1 | 14:39 | 31.0 |
| 16:30 | −12.0 | 54.2 | 54.3 | 44.9 | 33.4 | 40.5 | 58.3 | 47.8 | 46.3 | 43.3 | 47.2 | 32.3 | 36.1 | 17:42 | 29.1 |
| 17:00 | −12.3 | 50.0 | 50.5 | 42.3 | 31.3 | 36.7 | 54.8 | 44.3 | 43.5 | 40.1 | 43.6 | 30.6 | 35.0 | 15:24 | 27.7 |
| 17:30 | −12.7 | 44.0 | 46.5 | 39.8 | 30.0 | 33.4 | 50.8 | 41.5 | 40.6 | 35.9 | 39.0 | 28.7 | 31.8 | 20:26 | 26.7 |
| 18:00 | −13.6 | 38.6 | 41.5 | 36.9 | 27.8 | 28.8 | 45.7 | 37.2 | 37.7 | 31.5 | 33.2 | 26.1 | 27.8 | 9:00 | 24.8 |
| 18:30 | −14.2 | 32.7 | 36.7 | 33.5 | 25.8 | 24.4 | 40.5 | 33.5 | 34.5 | 26.8 | 27.5 | 23.4 | 23.4 | 11:40 | 23.0 |
| 19:00 | −14.5 | 26.7 | 31.2 | 30.1 | 23.8 | 20.4 | 35.7 | 29.6 | 31.3 | 22.5 | 22.0 | 20.1 | 20.6 | 4:34 | 20.8 |
| 19:30 | −14.9 | 23.8 | 27.7 | 27.1 | 22.1 | 17.7 | 32.2 | 26.7 | 28.5 | 20.4 | 19.6 | 18.5 | 19.1 | 12:01 | 19.9 |
| 20:00 | −14.6 | 21.9 | 26.0 | 25.3 | 21.0 | 16.7 | 29.4 | 24.7 | 26.6 | 19.2 | 18.2 | 17.8 | 18.2 | 21:17 | 19.2 |
| 20:30 | −13.6 | 20.9 | 24.6 | 23.8 | 20.0 | 16.6 | 27.3 | 23.6 | 25.1 | 18.7 | 17.4 | 17.2 | 18.1 | 22:49 | 19.4 |
| 21:00 | −13.1 | 20.2 | 23.5 | 22.5 | 19.6 | 15.9 | 26.0 | 22.4 | 23.8 | 18.3 | 17.2 | 17.0 | 17.4 | 12:43 | 18.9 |

**Table A1.** *Cont.*

| Key | |
|---|---|
| Artificial grass | A |
| Asphalt | B |
| Concrete slab (x40 mm) | C |
| Concrete slab (x80 mm) | D |
| Crushed rock (white) | E |
| Decking | F |
| Limestone block | G |
| Pavers (grey) | H |
| Pavers (red) | I |
| Pavers (sandstone) | J |
| Pine bark mulch | K |
| Polished stones (black) | L |
| Sand (grey) | M |
| Sand (white) | N |
| Shade cloth (black) | O |
| Shade cloth (cream) | P |
| Shade cloth (white) | Q |
| Soil (dry) | R |
| Soil (moist) | S |
| Turf grass | T |
| Ground cover (non-native, petunia) | U |
| Seedlings (native, saltbush) | V |

## Appendix B

**Table A2.** Average and maximum 30 min apparent minus ambient temperature calculated over the total measurement period at an emissivity of 0.95. The LEs are listed by summer average values in order from lowest to highest.

| LE | $\Delta T_{av}$ and Maximum, 30 min ($T_{app}$—$T_{amb}$) (Emissivity = 0.95) | | | | | | | |
|---|---|---|---|---|---|---|---|---|
| | Winter 2016 | | Spring 2016 | | Summer 2018 | | Autumn 2017 | |
| | Average | Max | Average | Max | Average | Max | Average | Max |
| Shade cloth (white) | −7.1 | −2.0 | −3.5 | 3.0 | −0.2 | 7.0 | −4.6 | 4.0 |
| Sand (white) | −5.9 | 0.0 | −1.0 | 11.0 | 2.1 | 11.0 | −3.4 | 5.0 |
| Crushed rock (white) | −4.5 | 1.0 | −2.5 | 6.0 | 2.5 | 10.0 | −2.4 | 4.0 |
| Limestone block | −2.4 | 4.0 | 0.1 | 8.0 | 2.7 | 8.0 | −0.6 | 8.0 |
| Turf grass | −1.5 | 7.0 | 4.1 | 14.0 | 4.3 | 14.0 | 0.8 | 14.0 |
| Shade cloth (cream) | −3.0 | 10.0 | 4.8 | 26.0 | 8.1 | 27.0 | 1.8 | 20.0 |
| Decking | −1.3 | 11.0 | 5.2 | 23.0 | 9.0 | 26.0 | 2.2 | 17.0 |
| Concrete slab (x40mm) | −0.7 | 10.0 | 6.4 | 22.0 | 9.7 | 24.0 | 3.4 | 20.0 |
| Concrete slab (x80mm) | 0.1 | 9.0 | 6.9 | 21.0 | 10.0 | 22.0 | 3.7 | 17.0 |
| Pavers (sandstone) | −1.1 | 7.0 | 5.6 | 18.0 | 10.1 | 24.0 | 2.8 | 16.0 |
| Soil (moist) | 0.2 | 12.0 | | | 11.8 | 32.0 | 4.7 | 23.0 |
| Pavers (red) | 1.2 | 13.0 | 5.9 | 20.0 | 13.4 | 32.0 | 5.1 | 22.0 |
| Pine bark mulch | 0.2 | 12.0 | 9.5 | 32.0 | 13.7 | 36.0 | 5.2 | 22.0 |
| Sand (grey) | 0.8 | 15.0 | 10.6 | 33.0 | 14.1 | 36.0 | 6.3 | 26.0 |
| Polished stones (black) | 1.4 | 13.0 | 8.1 | 26.0 | 14.6 | 32.0 | 7.5 | 24.0 |
| Artificial grass | 1.8 | 19.0 | 11.8 | 40.0 | 15.6 | 44.0 | 7.4 | 32.0 |
| Asphalt | 3.1 | 16.0 | 14.3 | 37.0 | 17.7 | 40.0 | 10.5 | 33.0 |
| Soil (dry) | 0.2 | 12.0 | 14.3 | 44.0 | 18.1 | 45.0 | 9.4 | 34.0 |
| Pavers (grey) | 4.2 | 19.0 | 13.0 | 34.0 | 18.3 | 38.0 | 8.6 | 29.0 |
| Shade cloth (black) | 4.6 | 26.0 | 15.5 | 48.0 | 19.6 | 49.0 | 10.7 | 39.0 |

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
