# Peer review of "Seasonal and Diurnal Surface Temperatures of Urban Landscape Elements"

_sustainability, doi:10.3390/su11195280_

Round 1

Reviewer 1 Report

The paper presents indisputable results regarding the contribution of different outdoor surfaces for the creation of Urban Heat Islands.

These results have multiple relevance with reference to: the knowledge of landscape designers; the creation and management of local planning regulations; the references for the environmental impact assessment.

The paper is well structured and very accurate; the data was collected with scientific rigor and the method is consistent and effective.
the paper can be accepted in the present form.

Reviewer 2 Report

Dear Authors,

This article focus on very interesting topic for architects and urban planners, providing relevant elements regarding the thermal behavior landscape elements (LEs).

The article presents a good quality in its writing.

Although the methodology is interesting considering its empirical nature, questions can be made regarding the consistency of the applied methods.

Results are relevant, however they must be considered for a particular location and limited weather conditions, nonetheless it adds to current knowledge on this particular topic.

Abstract

The abstract is consistent with the content of the article.

Introduction

This article provides an interesting introduction.

On the end of the introduction, the Authors should present a clear definition of the climate classification of the site in which the study was carried out. The Koppen-Geiger classification would be helpful, as it provides a reference regarding local climate characteristics.

Methodology

Please provide further information regarding the technical specifications of meteorological sensors and thermal camera, namely by identifying the name of the manufacturer and the model used.

This article presents a valuable estimation of the data error, thus allowing for a more rigorous analysis. 

The study presents several limitations, some of which are acknowledge in this article. The differences in surface deployment conditions for the two studies and the effect of wind, may have significantly affected the results.

Results

Some of the results were obtained under gentle to moderate wind conditions (winter and spring) in contrast with more calm occasions; these conditions may affect the results, as they will change the convection on the different study occasions. One of the assumptions of this studies is “linear for wind speeds under 5 m·s-1”, however winds were higher than this value during some of the measurements. Please provide a comment or an explanation on the possible implications of this circumstances.

Consider the application of statistical analysis to inform on the significance of the differences found in the data provided in this study.

Table 4 gives a false idea of linearity in the scale of thermal behaviour by LEs. Please consider change the representation to express the different levels of temperature interference through a proportional scale representation.

Conclusions

Conclusions were written in an unconventional manner, mostly by listing the outcomes of the study.

During the introduction, discussion and the conclusions mention should be made that the results are valid to a particular layout and climate conditions, and may not apply to a different context.

Detailed corrections

Line 73 - Please aggregate numbers [14, 15, 16, ...] etc.

In several lines, there the variables are omitted:

Lines 482-483 -  “asphalt and all the pavers are markedly higher for the larger sized LEs.” Please had the variable (EveΔTav)

Lines 514-515 – “With higher density there would be increased thermal mass and the evening data for asphalt would likely be much higher, perhaps comparable to grey pavers.”  Please had the variable

For the overmentioned comments I am suggesting that this article may be accepted after major revision.

Kind Regards  

Reviewer 3 Report

This paper investigates the seasonal and diurnal surface temperature of different materials or the landscape elements. Before its acceptance, authors should address following problems:

Major:

Abstract should be rewritten to clearly present the research content and results. Materials and methods must be restructured following: 2.1. materials; 2.2. field measurement settings; 2.3. Data analysis. The current version combines the materials and field measurement and presents them in phase 1 and phase 2. It is complex for readers to follow. Present the information of materials in either phase 1 or phase 2: albedo, size/dimensions, other physical properties.  Authors use the 24h delta T to present the all seasonal or daily surface temperature variations. Because of the missing nocturnal data, authors bring more errors into the realistic and robust data. Trying to just present the real data (you may divide them into diurnal and nocturnal section) of phase 1. Does the convective heat flux contribute to the data analysis? Results: Present the daily variation of weather conditions.  Results: Present the variation of all materials' surface temperature (each point every 30min) Shorten the conclusions, so long, seems you present the findings in detail again.  2.4 Limitations should be moved to the end of this paper.

Minor:

Line 21, the coolest

Line 55, a decrease in pervious surfaces , not just vegetation

Line 62, and [12]), add 'etc', or "other combinations"

Line 68, you may mean 'heat storage capacity', not the heat capacity. BTW, add references

Line 184, 'a worst case scenario for urban heat', not sure if it is the worst, be careful, as the temperature on that day was not the highest.

Round 2

Reviewer 2 Report

Dear Authors,

I am satisfied with the answers and changes introduced in this article, therefore I am recommending it for publishing.

Kind Regards,

Reviewer 3 Report

well done.